behaviour, evolution

social evolution, *Polistes* paper wasps, nest drifting, social network analyses

**Author for correspondence:**
Seirian Sumner
e-mail: s.sumner@ucl.ac.uk

†These authors contributed equally to the study.

‡Present address: Institute of Basic Biomedical Sciences (IBBS), B.S.R.C 'Alexander Fleming', Vari, Greece

¶These authors contributed equally to the study.

§Present address: Centre for Biodiversity and Environment Research, University College London Gower Street, London WC1E 6BT, UK.

# Multi-level social organization and nest-drifting behaviour in a eusocial insect

Thibault Lengronne[1,2,†], David Mlynski[4,†], Solenn Patalano[2,‡], Richard James[3,¶], Laurent Keller[1,¶] and Seirian Sumner[2,¶,§]

[1]Department of Ecology and Evolution, University of Lausanne, 1015 Lausanne, Switzerland
[2]Institute of Zoology, Zoological Society of London, Regent's Park, London NW1 4RY, UK
[3]Department of Physics and Centre for Networks and Collective Behaviour, and [4]Department of Biology and Biochemistry (plus CNCB), University of Bath, Bath BA2 7AY, UK

SS, 0000-0003-0213-2018

Stable social groups usually consist of families. However, recent studies have revealed higher level social structure, with interactions between family groups across different levels of social organization in multiple species. The explanations for why this apparently paradoxical behaviour arises appear to be varied and remain untested. Here, we use automated radio-tagging data from over 1000 wasps from 93 nests and social network analyses of over 30 000 nest visitation records to describe and explain interactions across levels of social organization in the eusocial paper wasp *Polistes canadensis*. We detected three levels of social organization (nest, aggregation and community) which exchange 'drifter' individuals within and between levels. The highest level (community) may be influenced by the patchiness of high-quality nesting habitats in which these insects exist. Networks of drifter movements were explained by the distance between nests, the group size of donor nests and the worker-to-brood ratios on donor and recipient nests. These findings provide some explanation for the multi-level social interactions, which may otherwise seem paradoxical. Fitness benefits across multiple levels of social organization should be considered when trying to understand animal societies.

## 1. Background

Eusociality is a widespread phenomenon in the animal kingdom and is considered as one of the major evolutionary transitions of life [1]. Eusociality in the insects—ants, termites, some bees and wasps—is one of the most complex examples of social organization in the animal kingdom [2]. Stable social groups are usually families, where non-reproductive members (workers) gain indirect fitness by helping reproductive members (queens) [3,4]. However, recent analyses suggest that higher level social structure, with repeated and frequent interactions between family groups, is common [5]. In some cases, workers appear to 'drift' to non-natal colonies where they behave as social parasites [6–8]; in other cases, drifters appear to perform helping behaviour [9], while in some species, no clear selective advantage for inter-nest drifting has been identified [10,11]. However, the lack of a detailed analysis of drifting patterns within and among populations is curtailing our ability to determine the importance of drifting as a social trait and understand why it evolves.

The paradox of inter-nest drifting by workers—whereby workers spend time in neighbouring non-natal nests—was highlighted by W.D. Hamilton as an anomaly to his theory on the genetical evolution of social behaviour [12]; he describes the 'transference of workers … to be not uncommon in some wasps' [12, p. 49] and suggests that it may be due to discrimination errors (perhaps owing to the putatively genetically viscous populations of clustered nests). He also compares the behaviour to that of sea birds in dense nesting aggregations, who will 'sometimes feed the hungriest chicks … rather than their own'

[12, p. 50]. Since Hamilton's observations, drifting behaviour has been reported in a number of *Polistes* wasps [9,13–16] and bees [5,7,8,10,11,17], but an understanding of the adaptive value of drifting has been hampered by a lack of experimental manipulation of the factors [18] that have been proposed to favour drifting. Even for *Polistes*—Hamilton's poster-child for drifting—our understanding is based on small-scale studies which were at best correlational in nature and did not account for the statistically challenging inter-dependency of the data [9,13–16].

To better understand the reasons underlying drifting and determine at what level of the population it occurs, we analysed thousands of nest visitation records and conducted experimental manipulations of potential drivers of drifting, in the tropical paper wasp *Polistes canadensis*. A previous study on this species found that greater than 50% of workers move between nests on which they appear to perform helping behaviour [9]; although this finding was remarkable, this previous study was based on a small sample size (157 tracked wasps), did not take account of the inter-dependency of the data and was purely correlational, lacking any manipulation of the conditions that may shape patterns of drifting. A recent study revealed that helping by drifters on non-natal nests could be explained by diminishing returns [19]. This study showed that theoretically, as the worker-to-brood ratio rises, the marginal productivity benefits of a worker's help diminishes; furthermore, empirical data for *P. canadensis* on productivity with respect to worker-to-brood ratios supported these predictions, providing a compelling explanation for the drifting paradox. Experimental manipulations of worker-to-brood ratios are required to test this explicitly.

Colonies of *P. canadensis* typically contain a single, active reproductive female, and, apart from drifters, the workers are usually the offspring of the queen and thus gain fitness benefits from helping on the natal nest [20]. Why a worker would spend time on any nest other than its own nest, to which it is highly related, remains a pertinent question. *Polistes canadensis* was in fact the subject of Hamilton's original observations on 'transference of workers' and thus is an excellent model within which to address outstanding questions on the significance of drifting in *Polistes*. Outstanding questions include whether these high-level social interactions are a general phenomenon that persists over time, whether drifting extends beyond the immediate vicinity of family groups and whether the observed inter-nest interactions are driven by adaptive traits. Our lack of understanding of inter-nest interactions is in part owing to the difficulty of tracking individual insects over significant time periods. Several tracking systems have now been designed to monitor continuous movements of multiple individuals and allow automated mapping of social interactions among groups [21–23]. Here, we analyse over 30 000 nest visitation records for over 1000 wasps, from 93 nests to measure the rate of interactions across multiple social levels in *P. canadensis* using Radio Frequency Identification (RFID). RFID tags facilitate the collection of precise, real-time quantitative information on the movement of individuals at different scales of biological organization [9,23,24]. In an attempt to identify correlates of inter-nest interactions and determine levels of structuring beyond the nest level, we analysed the drifting networks using statistical models adapted from those used to analyse human and non-human social networks (see for example [25,26]). We also experimentally manipulated helping pay-offs of specific nests to further identify the factors driving individuals' decisions and investigate the possible adaptive significance of spatially structured, multi-level interactions.

## 2. Methods

### (a) Network interaction and levels of social organization in *Polistes canadensis* populations

The field sites consisted of 93 small- to medium-sized post-emergence nests studied in three populations across several years in abandoned buildings near Panamá City and Colón, Panama. In Panama, nests of this species are rarely found alone; they tend to be spatially discrete, in patchily distributed aggregations. This applies to populations in natural substrates (e.g. trees, caves) as well as anthropogenic substrates like buildings [16].

*Polistes* nests lack an envelope, and so unlike other wasps, ants and bees, they have no delimited nest entrance (figure 1). To maximally capture wasp activity, the number of approach angles to the nest was restricted by fixing acetate sheets around the back of the nest. Wasps could thus enter and leave the nest only via the front (open-cell side). Two to four antennae (dependent on nest size) were then spaced across the front of the nest such that at least 80% of the accessible nest area was included in the detection zone [27]. Given that wasps could theoretically be recorded on either entry or exit of a nest, the 80% detection zone would result in an estimated 96% detection rate (using binomial probability). All wasps (with the exception of the queen, who was identified prior to tagging by observing egg-laying after the experimental removal of an egg) in these nests were captured and fitted with RFID tags (methods follow [9]). The RFID equipment consisted of passive tags (GiS TS-Q5Bee Tags), which code unique identification numbers; 3 cm diameter circular antennae (GiS TS-A37) that detected tagged wasps passing within a 3 cm radius; and readers (GiS TS-R64) which stored the date, time and unique identification number of each wasp tag as it passed within the detection range of the antenna. To avoid double-counting events for any single wasps (e.g. if they fly on/off the nest several times as they arrive/leave), we counted multiple registrations of any one tag within a 60 s period as a single event. Continuous automated RFID monitoring was conducted from 8.00 to 18.00 hours (the main foraging period) for each day of each study period.

Censuses of the numbers of wasps were performed every 3 days at night to estimate the group size (number of adults), and brood were mapped once a week to estimate the number (and developmental stage) of brood; these were used to provide a weekly estimate of nest size, and worker : brood ratio for each nest in each aggregation (defined as a collection of nests that exchanged wasps). Wasps from both sites (Panamá City and Colón) were collected at the end of the experiment for molecular analyses (except for 2010). Samples from 2009 were genotyped for estimating relatedness using seven specific microsatellite markers (Pcan01, Pcan05, Pcan09, Pcan15, Pcan16, Pcan23 and Pcan24 [27]). Samples from 2005 were genotyped as reported in [9]. Relatedness was calculated by using the program RELATEDNESS 5.0.8 and weighting nests equally [28]. Standard errors were estimated by jackknifing over loci.

Collected drifters showed no ovarian development from the 2005 population [9], suggesting that helpers are not hopeful egg-layers or social parasites. Drifters collected in 2009 could not be dissected because of tissue storage issues. Drifters were not collected in the 2010 experiment.

### (b) Experimental perturbation of the aggregation-level interactions

Experimental manipulations of the nests were carried out to test whether wasps respond to the changing needs of the nests. The three likely variables were the distance between nests, group size

**Figure 1.** Four levels of organization at which social interactions occur in populations of *P. canadensis*. (*a*) RFID tags allow individual identification of wasps as they pass in the detection zone of an antenna at each nest ((*b*) nest). The identity and frequency of tags (wasps) detected at each nest (node) determines the presence and weight (thickness) of 'edges' in the network ((*c*) aggregation). Wasps were detected moving between aggregations, via weak links between buildings in 2005 ((*d*) community, (ii)) and 2009–2010 ((*d*) community, (i)). Distances between buildings ranged from 15 m (2009C1–2009C2) to 750 m (2009C1 to 2009C4/2010MH). (Online version in colour.)

(number of wasps) and brood number (based on our pre-manipulation data analyses). It is impossible to re-locate nests with wasps within the home range of a nest, as any re-located wasps will re-orientate back to their original nest location (S. Sumner 2009, personal observation). We therefore focused on manipulating the ratio of wasp number (female adults) to brood number. To decrease the worker : brood ratio (and thus increase the need for help), we permanently removed 30% of the foragers on 14 nests (three nests in 2009 (one in 2009C1, one in 2009C2 and one in 2009C3) and 11 in 2010 (six in 2010SF and five in 2010MH)). Removing foragers is unlikely to upset the social dynamics in this species as the mortality rate for foragers is high [9]; equally, foragers tend to be low-ranked individuals who are unlikely to supercede the queen and so the manipulation is unlikely to change the outcome of queuing for dominance [29]. To increase the worker : brood ratio (and thus decrease the need for help), we permanently removed 30% of the brood on nine nests (three nests in 2009 (one in 2009C1, one in 2009C2 and one in 2009C3) and six in 2010 (six in 2010MH)). Large brood (at least 60% of medium and large larvae) were preferentially removed as they represent the most valuable brood and require the greatest helping effort to rear. Cells that had contained the removed brood were also removed to prevent wasps perceiving empty cells as a decrease in queen or nest quality. None of the manipulated nests was subject to both treatments and remaining nests for each aggregation were left unmanipulated; however, because all nests within an aggregation are socially connected (see results), the relative needs of all nests were affected by these manipulations, irrespective of whether they were physically manipulated by us. Thus, we were interested in whether there was a general network-level response to the changes in need, rather than specific paired responses of wasps or nests pre- and post-manipulation. We therefore treated the pre-manipulation data and post-manipulation data as separate, unpaired analyses (see next section).

## (c) Data analyses
The nest on which the wasp was originally tagged was assumed to be the natal (or 'home') nest. This assumption is supported by data from [9], which showed that workers appear to return to their natal

nests at night; wasps were collected directly off their nests before dawn, and so the nest of tagging was the likely home nest. For subsequent analyses, the nests on which a wasp was tagged is the 'donor' nest and any other nests visited are 'recipient' nests. Each nest was denoted a 'node' in a network, and the levels of drifting between the nests correspond to the 'edges' (connecting lines) between nodes (figure 1) with line thickness representative of the frequency of drifting. Arrows on the edges point from the donor to the recipient nest (figure 2). Network visualizations, depicting drifting between nests, were generated using the 'igraph' package [31] in R [32] and NETDRAW [33].

In 2005, nests from different buildings were monitored simultaneously. The 2005 drifting network (figure 2*a*) is clearly partitioned into communities on the basis of the buildings the nests were in. This partitioning was tested statistically using Newman's assortativity coefficient $r$ [34] (using the R package 'igraph' [31]) as a measure of the amount of within-building relative to between-building drifting. For this test, we used a binarized version of the network, i.e. drifting did (1) or did not (0) occur. The statistical significance of the partition by the building was determined by jackknifing the drifting network, as described in [34] to provide an estimate of the standard deviation on the values of $r$. The assortativity test was also conducted on the 2009 data (figure 2*b*), where nests from different buildings were not monitored simultaneously, but had periods of 1–2 weeks of overlapping monitoring. For the 2010 data, it was not possible to examine between-aggregation drifting since both 2010SF and 2010MH were monitored at two consecutive, but non-overlapping, time periods.

To examine the heterogeneity of the drifting networks between the $N$ nests in a given aggregation (and year), we computed a coefficient of variation $S$ [35] in the counts $O_{ij}$ of the number of unique wasps observed drifting from nest $i$ to nest $j$, $(i \neq j)$, relative to a null expectation $E_{ij}$. We employed a diffusive null model, in which wasps are more likely to drift to nests that are nearer, according to a Gaussian distribution of inter-nest distance $X_{ij}$ with mean zero and standard deviation $\sigma$. The count of wasps drifting *from* each nest, $R_i = \sum_{j=1}^{N} O_{ij}$, was preserved in the null, to control for the tendency of more populous nests to

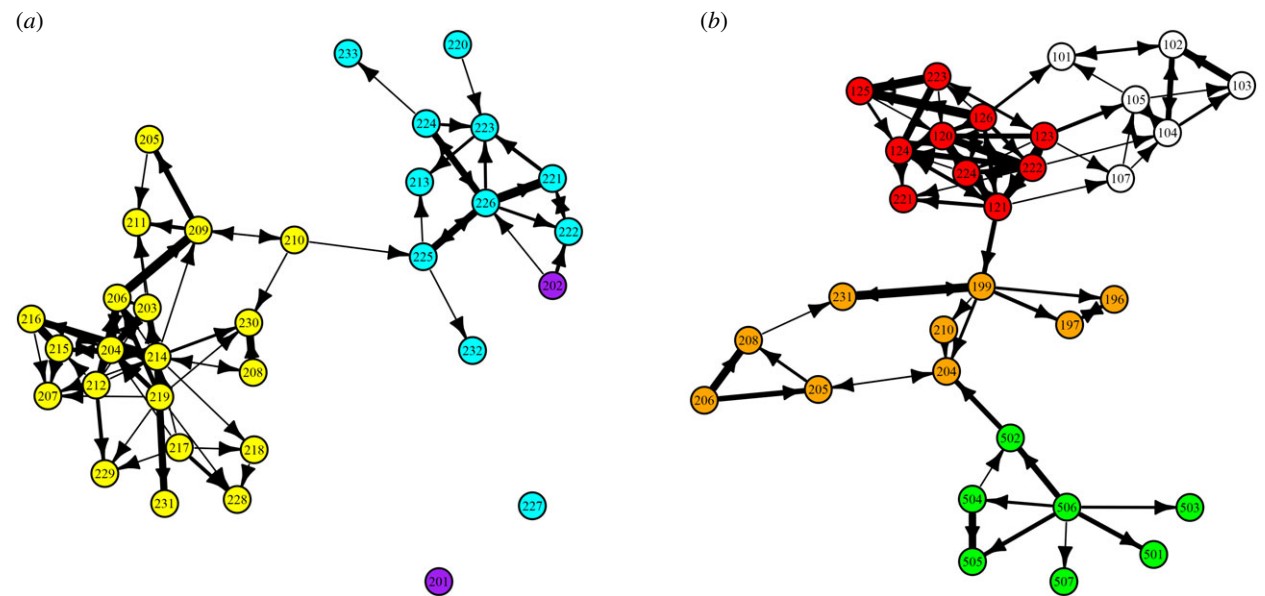

**Figure 2.** Networks depicting the drifting events recorded among nests and aggregations in 2005 (*a*) and 2009 (*b*) nests. Nodes represent nests and the edges (connecting lines) between them represent drifting events. Nodes are coloured to reflect the building the nest was in and the thickness of edges was weighted to represent the number of drifting events recorded. The network layout was determined using the Fruchterman–Reingold algorithm [30]. (Online version in colour.)

send more drifters ($r_{\text{Spearman's}} = 0.33$, $p_{\text{perm}} = 0.0008$). The expected number of drifting wasps is then:

$$E_{ij} = \frac{R_i \exp\left(-X_{ij}/2\sigma^2\right)}{\sum_{k=1}^{N} \exp\left(-X_{ik}/2\sigma^2\right)},$$

where $k$ is another nest index.

The variance in edge weights is

$$v = \frac{\sum_{i=1}^{N} \sum_{j=1}^{N} (O_{ij} - E_{ij})^2}{N(N-1)},$$

their mean is $\mu = \sum_{i=1}^{N} R_i / N$, and $S = \sqrt{v}/\mu$. To test the null hypothesis that drifting within a particular aggregation is no more heterogeneous than we would expect (given our null), we generated 9999 randomized versions of the drifting counts $O_{ij}$, in which we redistributed the observed number of wasps leaving each nest, $R_i$, among the other $N-1$ nests, using the same Gaussian tendency for wasps to drift to nearer nests. We calculated $S$ for each randomization, to generate a distribution of $S$ under our null. In all these calculations, the standard deviation of our Gaussian function, $\sigma$, is a free parameter; very large values (compared with typical inter-nest distances) give a 'flat' null, in which wasps are equally likely to drift to all other nests in the aggregation; very small values give a null in which wasps drift almost exclusively to the nearest nest. We varied $\sigma$ as a free parameter and conservatively chose the value $\sigma^*$ that gave the most support for acceptance of the null hypothesis. Finally, we used a combination test [36] to assess the probability that, overall, the heterogeneity in the wasp drifting network can be explained by our diffusive null.

Consistent patterns of drifting between nests over time (persistent edges) could be indicative of stable factors that regulate drifting, suggesting that the snapshots of these systems that we have observed are generalizable. To identify if there were fewer transient and a greater number of consistent or recurring edges than we would expect if wasps drifted without preference, the number of edges that were present between the same nests for greater or equal to one, two, three and all four of the observation periods were measured. Data collected from the 2005 population were used, which consisted of two main nest aggregations (2005S1, 2005S2) monitored simultaneously for drifting over four consecutive periods of 5 days.

We used a null model to test whether the number of edges recorded was statistically different from expectation. For each observation period, edges were randomly allocated between nests, preserving the total number of edges found within and between aggregations, while also preserving the number of recipient nests for each donor nest. The number of transient and recurring edges in each resulting null network was measured. This process was repeated 10 000 times to generate a distribution of null frequencies for each consistency threshold with which the observed figures could be assessed for significance. A p-value was derived based on the proportion of null networks which had an equal or more extreme number of edges at the given threshold than the observed network.

## (d) Variables that explain the networks

The size of individual nests combined with the distance, relatedness and difference in worker : brood ratios between nests were suspected to influence patterns of drifting between nests. The number of adult female wasps which belonged to a given nest was taken as its size. The difference in worker : brood ratios between nests was calculated as the difference in the number of workers divided by the number of brood for each nest in the dyad—this was a directional measure. Relatedness data was not available for the 2010 dataset. As missing data poses a considerable problem for many network modelling approaches [37], the distance was used as a proxy for relatedness given both the significant inverse correlation between the two ($r_{\text{Spearman's}} = -0.554$, $p = 0.001$, $n = 290$) and previous findings [9]. This correlation was identified by comparing the relatedness and distance between pairs of nests for which there was complete data. To determine the significance of the coefficient, the relatedness values between nests were permuted within each aggregation, holding the position of nests constant; this was repeated 4999 times. The distribution of null correlation coefficients produced as the product of this null model was used to determine significance. Most of the closely related nests were situated within 1 m of each other (median relatedness = 0.20025, $n = 132$), with nests with a distance greater than 1 m being significantly (Wilcoxon $W = 16610$, mean null $W = 10$ 439.15, $p < 0.0002$) less closely related to each other (median relatedness = $-0.051214$, $n = 158$).

To identify the factors which served to be significant predictors of drifting, a logistic multiple regression quadratic assignment

procedure (MRQAP) was used [38,39] where the number of drifting events between pairs of nests, e.g. nest A to nest B (designated as a donor (nest A) and a recipient (nest B)) was the response variable, and distance, nest size and worker : brood ratio the explanatory variables of interest. To accurately account for contribution to foraging effort, we used the counts of medium and large larvae which are fed the bulk of the forage. Owing to the sparseness of drifting data (i.e. many zeros), and the little variation in non-zero numbers, for this analysis, we binarized the data and used a logistic model. The effectiveness of the MRQAP has been shown to degrade under conditions of extreme skewness [40]. To control for an identified tendency for larger nests to send more drifters, the nest size of both the donor and recipient nest was included as explanatory variables in the model. Owing to the various kinds of dependency inherent in network data [26], the significance of each individual factor and the model as a whole were determined using a randomization procedure. Owing to a slight correlation between a nest's size and its difference in worker : brood ratio with other nests (Spearman's $\rho = \pm 0.18$, $p = 0.0002$), the randomization procedure chosen was a double semi partialing quadratic assignment procedure (QAP-DSP), which has been demonstrated to be robust to the effects of autocorrelation and co-linearity when determining significance [40]. QAP-DSP was a departure from the out-strength constrained edge randomizations used thus far on the basis that the QAP-DSP approach is well established for use in modelling networks. Furthermore, including nest size as a variable instead of constraining for it in the randomization procedure allowed its effect on drifting to be characterized.

The same analysis was conducted on the data from the manipulation experiments (excluding the 2005 aggregations) as applied to the pre-manipulation data. It was not appropriate to compare directly the differences in drifting on nests before the manipulation and afterwards because (as we report above) all nests within a network are socially connected: this means that even though we manipulated only a portion of nests in each aggregation, the relative brood-care needs of all nests were affected, not just the ones we manipulated. As such, we treated the manipulated data as a replicate experiment of the pre-manipulation data to test whether wasps are able to respond to the changing brood-care needs within the networks generally.

The analysis was carried out using a modified version of the 'netlogit' function in the 'sna' [41] package in R [32]. The function was modified to include stratification, or a block structure; this was so that data from all aggregations across all years could be included into one model while partitioning the QAP-DSP to take place only within blocks that could exchange wasps and not across blocks that could not interchange wasps owing to substantial separation in time. Through including all aggregations in one model in this way, each aggregation was treated as independent from each other in some respect and the analysis was restricted just to look at drifting within-aggregation. This step was justified given the strong partitioning of the wasp drifting networks on the basis of building in the 2005 and 2009 data. However, one limitation of partitioning, the randomization approach underlying the model is that aggregation-wide effects cannot be tested.

### (e) Assortativity by building: 2009 null model and results

In contrast with 2005, nests were not monitored simultaneously in 2009. We used a null model to check our results were robust to the fact that drifting in certain directions could not occur between buildings that did not have overlapping monitoring periods. We compared the observed proportion of within-building drifting (Tr(e)) to the same measure under the null model. In each of 4999 null networks, the out-degree or number of edges leaving each nest was fixed, but the destinations of those edges were randomized within given constraints. These constraints were that edges can go both ways between C1 and C2 nests and C3 and C4 as these had periods of simultaneous monitoring. However, edges were only allowed to go from C1 or C2 to C3 or C4, as C3 and C4 were monitored later in time than C1 and C2 with no overlap, but wasps were still present with RFID tags from C1 and C2. C4 was monitored for a day when tagged wasps from C1 could have been present; given this, the large distance between these aggregations (750 m) and the absence of any observed drifters between these aggregations, no drifting events were randomly allocated between these aggregations. A $p$-value was calculated by examining the proportion of null networks with a trace higher than the observed network. The results are as follows: observed Tr(e) = 0.9125, $p = 0.0002$, mean of null Tr(e) 0.3367175, suggesting that within-building drifting differed from random.

## 3. Results

### (a) Network interaction and levels of social organization in *Polistes canadensis* populations

The 93 nests used in this study had a mean number of 19.6 ± 1.4 wasps per nest. Nests were clustered into three populations found in abandoned buildings (2005: $n = 33$ nests, three buildings (S1–S3)) as described in [9]; 8°54′44″ N, 79°33′47″ W), and Colón (2009: $n = 32$ nests, four buildings (C1–C4); 2010: $n = 28$ nests, two buildings (SF & MH)) as described in [20,27]; 9° 24′08.28″ N, 79°52′19.41″ W, Republic of Panamá (figure 1). Of the 1599 tagged wasps, 1009 were recorded at least once, generating a total of 30 249 records (one detected arrival/ departure of a wasp) of which 2563 (8.5%) were drifting (inter-nest movement) events. Each nest was denoted a 'node' in a network, and the levels of drifting between the nests correspond to the 'edges' (connecting lines) between nodes (figure 1).

On average 40.4 ± 3.9% of the individuals ($n = 403$ wasps, 93 nests) were drifters. The vast majority of nests (92.7 ± 0.1%) received or produced drifters: 85 ± 0.02% of the nests received (recipient nests) and 71.1 ± 0.3% of these nests produced (donor nests) drifters, indicating that drifting is a general phenomenon and not restricted to particular nests; 19.3% of nests showed extreme levels of drifting, with greater than 60% of wasp records pertaining to drifting events.

The network analyses revealed up to three levels of social organization (figure 1). The lowest level was the nest where workers are expected to remain faithful because they are more closely related to those brood ($r = 0.69 ± 0.02$, mean ± s.e.) than to brood from neighbouring nests ($r = 0.12 ± 0.01$; mean ± s.e.; $n = 63$ drifters, 129 pupae and 26 nests) [27]. Consistent with this view, a median of 96% of wasps leaving a nest returned to the same nest (in 75% of the studied nests, between 70% and 100% of departing wasps returned directly to the natal nest).

The second level of social organization was between nests belonging to the same aggregation, where aggregation is defined as a collection of nests exchanging wasps (figure 1). There were eight aggregations and the number of nests per aggregation ranged from 6 to 20 (11.3 ± 1.7, mean ± s.e.; figure 2). The drifting networks in 2005 and 2009 were explained significantly by these aggregations; moreover, there was a significant assortment with respect to building identity (Newman's assortativity coefficient $r$ [42] for study years 2005 ($r = 0.893$) and 2009 ($r = 0.867$; jackknifing revealed

**Table 1.** Variability of drifting networks. (Social differentiation (*S*) measures the variation in edge weights (denoting the number of drifting wasps in this case) in the drifting network of each aggregation, compared with a null in which wasps are more likely to drift to nearby nests. This tendency is modelled by a Gaussian function with standard deviation $\sigma$ and mean 0. For each aggregation, we tuned $\sigma$ to give the largest *p*-value for the hypothesis that the observed value of *S* could have occurred by chance, given the null model. 'Mean null *S*' is the mean social differentiation observed across 9999 null networks. For four of the eight aggregations (†), no value of $\sigma$ ever gives a null *S* greater than or equal to the observed value, so *p* < 0.0001; we chose $\sigma^* = 10.0$ as an indicative value for the purposes of presenting *S* and its mean null value.)

| aggregation | no. of nests | best fit null $\sigma^*$ (m) | mean null *S* | observed *S* | *p*-value |
|---|---|---|---|---|---|
| 2005S1 | 20 | 5.0 | 3.536 | 4.244 | 0.0295 |
| 2005S2 | 11 | 10.0† | 1.996 | 4.786 | <0.0001 |
| 2009C1 | 6 | 10.0† | 0.803 | 2.562 | <0.0001 |
| 2009C2 | 10 | 10.0† | 1.029 | 1.654 | <0.0001 |
| 2009C3 | 9 | 0.65 | 1.090 | 1.242 | 0.2687 |
| 2009C4 | 7 | 3.0 | 1.472 | 1.604 | 0.326 |
| 2010SF | 12 | 0.9 | 1.114 | 1.598 | 0.008 |
| 2010MH | 16 | 10.0† | 2.154 | 3.685 | <0.0001 |

assortativity values to be 15.1 and 17.8 s.d. away from 0 in 2005 and 2009, respectively; figure 2), suggesting that nests largely exchanged wasps with other nests in the same building. Structuring by drifting networks partitioned the data into eight different aggregations of nests: two aggregations in the 2005 population (2005S1 and 2005S2; 2005S3 had only two nests); four aggregations in the 2009 population (2009C1, 2009C2, 2009C3 and 2009C4); and a further two aggregations in 2010, (2010SF and 2010MH) (table 1). The mean distance between nests within an aggregation ($6.1 \pm 0.2$ m, mean $\pm$ s.e.) was significantly less than the distance between aggregations ($425 \pm 146$ m, mean $\pm$ s.e.; two-tailed unpaired *t*-test, $p = 0.02$).

The third level of social organization detected was between aggregations belonging to the same community where a community is defined as a collection of aggregations exchanging wasps (figure 1). Aggregations in two study periods were monitored simultaneously (for 2005 data) or with overlapping periods (for 2009 data—C1 and C2 were simultaneously monitored for 6 days; C3 and C4 for 5 days); 5 and 19 drifting events respectively occurred *between* aggregations (buildings) in these periods. In each case, these rare drifting events between nests were attributed to a single drifter which visited the same pair of nests, one in each of the two aggregations (i.e. unidirectional, un-reciprocated), albeit multiple times (2005: $1.7 \pm 0.7$ visits (1–3 visits per drifter); 2009: $3.0 \pm 0.9$ visits (1–6 visits per drifter)).

## (b) Inter-nest interactions may be non-random and persistent

Two lines of evidence suggest that drifting patterns are non-random. First, within seven of the eight aggregations, the patterns of drifting observed were more heterogeneous than expected if drifting was a random or a diffusive process. Specifically, the observed social differentiation *S* (a measure of how heterogeneous the observed drifting network is) was significantly greater than expected under a null model of random drifting with a bias to nearer nests (table 1; figure 2; see methods for details). Moreover, a combination test [36] showed that, overall, the heterogeneity in the wasp drifting networks can also not be explained by a diffusive null ($n =$ eight aggregations; $\sum -2 \ln p = 95.3$; $p < 0.005$), suggesting

that the non-random patterns of drifting are driven by variation in physical and biological factors other than distance.

The second line of evidence suggesting that drifting is non-random is that patterns of drifting between nests show temporal persistence. There were fewer transient and more persistent (recurring) links than we would expect if drifting were random or performed only fleetingly (table 2; see the electronic supplementary material). Moreover, persistent links were not between nests that were very close together (see the electronic supplementary material). Over the four consecutive monitoring periods, six pairs of nests had persistent edges (8.2% of observed edges remained over all four monitoring periods). We also found 12 edges (16% of observed edges) that occurred in at least three of the four 5-day monitoring periods and 22 edges (30% of observed edges) that occurred in at least two of the four monitoring periods. The remaining 51 edges were more transient with detections occurring during only one monitoring period. There were significantly fewer transient edges and significantly more recurring edges at each threshold than we would expect compared to null versions of the dataset (table 2). Persistent edges correspond to a higher level of drifting between nests than transient edges, with on average $4.3 \pm 0.7$ drifting events in any pair of nests (compared with $2.4 \pm 0.6$ for more transient edges; Mann–Whitney *U*, $W = 2236$, $p_{\text{perm}} < 0.0002$), and up to 31 drifting events for a pair of nests in a single monitoring period (max value = 31 drifting events for persistent edges and 18 drifting events for transient edges).

## (c) Biological traits explain patterns of inter-nest drifting

We identified three biological traits that may explain the patterns of drifting. The first trait was nest size (number of wasps): donor nests (where drifters originate from) were on average larger than recipient nests (which receive drifters) (table 3, estimate = 0.02; $p = 0.0012$). However, we detected no significant relationship between the size of the recipient nests and their tendencies to receive drifters (table 3, $p > 0.4$).

The second trait was the distance between nests: drifting was more likely between nests that were close together than far apart

**Table 2.** Drifting rates are consistent over time. (The mean value of edges which meet each threshold was taken from 4999 null networks and the standard deviation around the mean ($\sigma$) included. $p$-values state the probability that the null model would produce equal to or more edges meeting the given threshold, unless *, which indicated the probability of an equal number or fewer transient edges being produced by the null model.)

| number of observation periods | observed (cumulative) | mean null | $\sigma$ null | $p$-value |
|---|---|---|---|---|
| >=1 | 73* | 96.66 | 2.74 | <0.0002 |
| >=2 | 22 | 16.52 | 2.48 | 0.0284 |
| >=3 | 12 | 2.44 | 1.26 | <0.0002 |
| =4 | 6 | 0.17 | 0.41 | <0.0002 |

**Table 3.** The results of the logistic model on the pre-manipulation drifting data. (The estimated value for each coefficient is shown under 'estimate'; the exponent of the estimate is shown as Exp(b) showing how a unit change in the coefficient affects the odds of observing drifting. Pr(< = b) indicates the probability that the estimate of the coefficient is higher than expected under the null (i.e. higher levels of drifting than expected by the null); Pr(>=b) indicates the probability that the observed coefficient has an estimate lower than expected under the null hypothesis (i.e. lower levels of drifting than expected by the null). Significant results are highlighted in italics and discussed in the text; however, effect sizes for worker : brood ratios are very small (0.000 at 3 decimal places (d.p)), suggesting there may be little biological effect here. Effect sizes and probabilities suggest that wasps drift less as distance increases and as size of donor nest increases.)

| coefficients | estimate (3 d.p) | Exp(b) | higher levels of drifting than expected Pr(<=b) | lower levels of drifting than expected Pr(>=b) |
|---|---|---|---|---|
| (intercept) | −0.639 | 0.528 | >0.9998 | <0.0002 |
| distance | *−0.003* | 0.997 | *<0.0002* | >0.9998 |
| worker : brood ratio donor | 0.000 | 1.000 | 0.9930 | 0.0070 |
| worker : brood ratio recipient | −0.000 | 0.999 | 0.0086 | 0.9914 |
| donor nest size | *0.020* | 1.021 | 0.9988 | *0.0012* |
| recipient nest size | −0.013 | 0.987 | 0.5314 | 0.4686 |

(table 3, $p < 0.0002$), although the effect size was small (estimate = −0.003). Nests in aggregations show significant genetic viscosity such that nests close to each other are more closely related than nests further away [20,27]. In line with this, there was an inverse correlation between the distance between nests and their relatedness ($r_{Spearman's}$ = −0.554, $p_{perm}$ = 0.001, $n$ = 290). Wasps appear to be drifting to closely related nests ($r$(drifters to visited nests) = 0.17 ± 0.01; $n$ = 63 drifters; 129 brood), where the indirect fitness benefits of helping are greatest.

The third trait was worker : brood ratios. We predicted that nests with the lower worker : brood ratios would donate fewer drifters, but receive more drifters. Likewise, we predicted that nests with the higher worker : brood ratios would donate more drifters and receive fewer drifters. We found some support for both these effects in the unmanipulated nests (table 3), although effect sizes were small. Similarly, in the experiments where we manipulated the worker : brood ratios (see methods), a detectable effect of worker : brood ratio on drifting patterns was apparent in nest donating drifters, but not for the nests receiving nests, although again the effect sizes were small (estimate = −0.079; $p$ = 0.0058; table 4). Distance between nests remained a significant predictor of drifting (estimate = −0.002; $p$ = 0.019; table 4), but neither donor nor recipient nest size had a significant effect despite the fact they had been altered. Our analyses also show that drifters were significantly more closely related to brood on their natal nests ($r$ = 0.56 ± 0.02 s.e.) than those on the nests they drift to ($r$ = 0.17 ± 0.01 s.e.; $n$ = 63 drifters; $n$ = 129 brood; $p$ < 0.001, paired-$t$-test).

## 4. Discussion

Determining the extent, nature and biological importance of nest drifting in social insects has been an outstanding question since Hamilton first described the behaviour in tropical *Polistes* in his 1964 treatise on kin selection theory. Using over 30 000 records generated by an automated real-time monitoring system of over 1000 individually tagged paper wasps across 93 nests, we provide, to our knowledge, the most comprehensive analysis to date of inter-nest drifting in a social insect. We identified up to three levels of social organization above that of the individual: nest, aggregation and community (figure 1). We showed that high-level inter-actions (e.g. at levels above the family group/nest) are highly structured, non-random, and that they cannot be explained simply through a simple model of diffusion over distance. Finally, we identified three biological traits that may explain the patterns of drifting.

Some of the earliest *Polistes* researchers noticed nest drift-ing behaviour [12,13,43]; the behaviour appears to be prevalent across many species, especially those in tropical regions [9,14–16]. Our study now provides conclusive quan-titative evidence that high levels of inter-nest interactions are a mainstay of social dynamics across populations in *P. canadensis*. We show that on average 30% of workers drift, and in doing so create multiple layers of social organization: drifters connect groups of closely aggregated nests, and rare drifting events link aggregations within populations. Detec-tion of movements between aggregations was surprising as

**Table 4.** The results of the logistic model on the post-manipulation drifting data. (Significant results are highlighted in italics and discussed in the text, column headings as in table 3. Effect size and *p*-value suggest that wasps drift less as distance increases and when worker : brood ratio decreases on donor nests. The effect size was relatively large for the response to worker : brood ratios on recipient nests, although not significant.)

| coefficients | estimate | Exp(b) | higher levels of drifting than expected Pr(<=b) | lower levels of drifting than expected Pr(>=b) |
|---|---|---|---|---|
| (intercept) | −0.969 | 0.380 | <0.0002 | >0.9998 |
| *distance* | *−0.002* | 0.998 | *0.0188* | 0.9182 |
| *worker : brood ratio donor* | *−0.079* | 0.924 | 0.9942 | *0.0058* |
| worker : brood ratio recipient | 0.138 | 1.148 | 0.1806 | 0.8194 |
| donor nest size | 0.021 | 1.021 | 0.9182 | 0.0818 |
| recipient nest size | −0.021 | 0.979 | 0.0676 | 0.9324 |

relatedness decreases with distance [27]. *Polistes canadensis* are also found nesting in aggregations on natural substrates like trees and caves ([15] S. Sumner 2005, personal observation); although we did not find such aggregations near to our study sites, it is possible that there were other unmonitored aggregations nearby that were exchanging wasps with our study aggregations, meaning that the level of inter-aggregation drifting reported is actually underestimated. Future studies would benefit from intensive monitoring of nests in all detectable aggregations concurrently for longer periods of time (as in the 2005 dataset) to properly understand the importance of these apparent 'weak links'.

We identified three putative drivers of drifting patterns: nest size, nest proximity and worker : brood ratios. By drifting to smaller nests, nests which are close by (and thus tend to be more closely related), and by drifting from nests that have lots of workers relative to the brood number, inter-nest movements may have some adaptive significance. In the experiment where the conditions on the nests were manipulated, workers appeared to alter their drifting behaviour in response to reduced group size and increased need for help on their natal nest; however, the effect sizes were small. Moreover, the complexities of our manipulation may have disrupted the conditions of all nests in each aggregation: altering the conditions on more than one node (nest) in a network may alter inter-node interactions in complex ways that are hard to control for or detect. In our case, we tried to minimize this effect by only manipulating 30% of the nests in each experimental aggregation; however, this may still have been too much, leading to perturbation of the whole network of interactions between nests. With this caution, these experiments suggest that drifting is not simply a probability function of group size (i.e. big nests do not donate more drifters simply because they are big) but that drifting rates changed in response to brood removal even when the number of wasps (group size) remained unaltered. Our analyses also revealed that the size of donor nest may not be a significant explanatory variable after manipulation, even though there were fewer wasps: in other words, drifting patterns did not respond to manipulation of group size per se, only the ratio of group size to brood (table 4; see also the electronic supplementary material). That larger nests donate more drifters, and smaller nests receive more drifters makes sense since the added value of each worker diminishes with increasing group size in many eusocial insects [44,45], including our study species [9]. A recent theoretical paper concluded that

such diminishing returns for helpers can, theoretically, explain drifting in insects like *Polistes*; moreover, this study showed for our study species (*P. canadensis*) that productivity pay-offs diminish as worker : brood ratio increases, providing the fitness landscape for drifting behaviour to be adaptive [19]. Our findings provide, to our knowledge, a first experimental test of these empirical patterns and theoretical predictions.

In conclusion, our analyses expose the complexities of the drifting phenomenon and identify some of the biological and physical factors that contribute to heterogeneity in drifting. Wasps appear able to respond to changes in the conditions of their social environment in a way suggesting that drifting might be at least partly adaptive. The fitness pay-offs of such a strategy across multiple levels of social organization have been largely overlooked in previous studies on proximate and ultimate drivers of animal group living. Advances in automated monitoring methods and quantitative statistical analyses that take account of the non-independence of social interaction data, as used in our study, are likely to bring us closer to a more comprehensive understanding of the complexities of social behaviour and its evolution.

**Ethics.** This work was conducted under fieldwork permits no. SE/A-33-09, no. SE/A-65-10; no. SEX/A-44-10, granted by the Panamanian authority ANAM.

**Data accessibility.** Data are available from the Dryad Digital Repository: https://doi.org/10.5061/dryad.v41ns1rvc [46]. Dataset 1: RFID monitoring data for 2005, 2009, 2010. Raw data provided, with wasp identification, nest, site, year. Dataset 2: genotyping and relatedness data for wasps collected in 2009. Dataset 3: matrices used in social network analyses. The data are provided in the electronic supplementary material.

**Authors' Contributions.** S.S. and L.K. designed the project. T.L. and S.P. conducted fieldwork. T.L. and D.M. conducted analyses. R.J. supervised and conducted analyses. S.S., R.J. and L.K. wrote the manuscript. All authors approved the manuscript. All authors gave final approval for publication and agreed to be held accountable for the work performed therein.

**Competing interests.** Authors have no competing interests

**Funding.** This study was funded by NERC grant NE/G000638/1 (S.S.), several grants from the Swiss NSF and an ERC advanced grant (L.K.) and a BBSRC studentship (D.M.).

**Acknowledgements.** We thank W. Wcislo and M.J. West-Eberhard for their hospitality at the Smithsonian Tropical Research Institute (STRI), Panama, and all the staff at the Galeta Field Station (STRI), especially J. Morales, I. Grenald and S. Heckleton. We thank N.J.B Isaac for help with R coding. The manuscript was greatly improved by the comments of three anonymous reviewers, the Associate Editor and the Editor.

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
