## [Peer Review File · Proceedings of the Royal Society B: Biological Sciences]

Review History

RSPB-2020-0999.R0 (Original submission)

Review form: Reviewer 1

Recommendation

Accept with minor revision (please list in comments)

Scientific importance: Is the manuscript an original and important contribution to its field?

Excellent

General interest: Is the paper of sufficient general interest?

Excellent

Quality of the paper: Is the overall quality of the paper suitable?

Excellent

Is the length of the paper justified?

Yes

Should the paper be seen by a specialist statistical reviewer?

No

Do you have any concerns about statistical analyses in this paper? If so, please specify them explicitly in your report.

No

It is a condition of publication that authors make their supporting data, code and materials available - either as supplementary material or hosted in an external repository. Please rate, if applicable, the supporting data on the following criteria.

Is it accessible?

No

Is it clear?

N/A

Is it adequate?

N/A

Do you have any ethical concerns with this paper?

No

Comments to the Author

This paper addresses an interesting question, namely, how is drifting behaviour between eusocial wasp nests is affected by the needs of those nests, their size and their spatial distribution. A manipulation of worker:brood ratio was used to change the need of the nests. The distance between nests and worker population were also taken into account as explanatory variables, with distance being used as a proxy for relatedness. The methods are somewhat complex, both because of the nature of the network analysis, and because the authors have combined multiple datasets collected over several years in this paper. Despite these challenges, I view the experiment as being well-executed and the paper is very well-written. The results are interesting, not just to wasp biologists, but to those working on cooperation and social behaviour more broadly.

I have no significant concerns about the work or its presentation here. I make a few minor comments below on issues of clarity in the manuscript. These will be easy for the authors to implement, and the resulting ms will be high quality.

The methods sections contain quite a lot of results, but I don't think that is a problem – they are needed to explain the decision made in the methods, and it is clearer to include them in the methods than to make the reader track back and forth between the results and the methods to understand the rationale for the approaches taken. I am less keen on the mixing in of discussion with the results, e.g. lines 407-410 are a somewhat speculative hypothesis, and while they are consistent with the data, I don't think they merit a place in the results section. Similarly lines 420-422 of the results belong in the discussion. Lines 427-430 of the results are methods, and repeat what has already been said, so could be deleted.

When the methods are introduced, the number of wasps (presumably adults) per nest is given, but it would be helpful to the reader to state whether each nest has a single queen, or whether multiple females are reproductively active. This information is provided implicitly later, but it would be helpful to give it explicitly up front, because this aspect of ecology is variable within *Polistes*, and is highly relevant to the implications of drifting

Lines 163 and 281. I suggest a qualifier preceding 'connected', e.g. socially connected, because readers who are not familiar with paper wasp biology could assume this means structurally connected.

Line 224. I think the phrasing is a bit confusing here. It isn't the significance of the number of edges that is being determined, but whether the number of edges differs statistically significantly from the null model. Whether the number of edges is significant is a broader question (and an interesting one, but not addressed here)

Line 286. Is the correlation here also a Spearman's rho?

I think a supplementary table summarising the key differences between the datasets, and which datasets were used for which parts of the analyses would be useful – this information is given in the text, but it is hard to keep track of, and very hard to assess the implications of this, because there are so many exceptions throughout.

I wonder whether the authors still think, after their analysis, that 'drifting' is the correct term for these inter-nest movement events? Drifting rather implies an aimless unintentional behaviour, and the results don't seem consistent with this idea. I don't suggest the authors should change this throughout – that would be confusing with regard to previous literature. But perhaps they might consider raising an alternative term for this behaviour in their discussion? Or at least comment on the likely unfittingness of the current term? But I leave that to the author's discretion, and realise they are likely up against the word count, given the complexity of their methods anyway.

Review form: Reviewer 2

Recommendation

Reject – article is not of sufficient interest (we will consider a transfer to another journal)

Scientific importance: Is the manuscript an original and important contribution to its field?

Good

General interest: Is the paper of sufficient general interest?

Acceptable

Quality of the paper: Is the overall quality of the paper suitable?

Good

Is the length of the paper justified?

Yes

Should the paper be seen by a specialist statistical reviewer?

No

Do you have any concerns about statistical analyses in this paper? If so, please specify them explicitly in your report.

No

It is a condition of publication that authors make their supporting data, code and materials available - either as supplementary material or hosted in an external repository. Please rate, if applicable, the supporting data on the following criteria.

Is it accessible?

Yes

Is it clear?

Yes

Is it adequate?

Yes

Do you have any ethical concerns with this paper?

No

Comments to the Author

This paper used RFID-tag data together with nest demography and relatedness data to describe the movement patterns of individuals in a *Polistes* wasp. As *Polistes* wasps are one of the key taxa in our current understanding of primitively eusocial insect societies, this paper has a clear target audience in social insect researchers. The paper also introduces statistical rigor into the analyses of drifting patterns through network statistical analyses.

The main results of the paper are that, first, drifting between nests is common, second, that drifting is not random and not explained by spatial proximity alone, and third, that most movement between nests take place locally and between location drifting is rare. Furthermore, larger nests seem to send out more drifters, and also the worker to brood ratio in a nest explains some variation in drifting patterns. Given that the role of relatedness underlying drifting patterns has been described in this species earlier, and some of my concerns I detail below that undermine the robustness of some interpretations, I feel that none of these results have the broad general appeal that I would expect from a paper in this journal, and thus I see this paper more fitting in a more specialist animal behaviour or behavioural ecology journal.

Apart from the comments that follow, the data collection methods and statistical analyses are thorough, and the paper is well written and clearly argued.

Methods:

1. How was it determined which is the “home nest” of an individual from which it drifts to other colonies? I assume this is straightforward (the nest where they were originally captured?), but should be described anyway, as this is important for the speculation about indirect fitness gains: it makes a difference whether individuals from a big nest drift to help at a small nest where help is more valuable, or vice versa.
2. I’m not sure I follow the logic of lines 122-125. Why is the potential source of error smaller if only unique drifting wasps are used, can’t they be misallocated as well? Also, are these unique drifting wasps on a given observation day, or over the whole time period? If the latter, some of the results are hard to understand: lines 366-368.
3. A lot of the results are written in a way that suggests that the data allowed differentiating between a wasp entering a nest and leaving a nest (rather than just being detected at the reader), but the methods don’t give this picture, how was this done?
4. It would be nice to have some indication of how common nests of the species are outside the studied locations. If there are a lot of nests not observed, it makes a difference on the interpretation of the larger spatial scale results. Of course it is not feasible to observe every single nest within flight radius and I’m not expecting that, but this should be at least briefly discussed. This is also relevant for the speculation on the effects of anthropogenic change in the discussion.
5. the methods also contain a lot of analysis results, that should perhaps be moved to the Results section.

Results

6. Out of 1599 tagged wasps 1009 were observed again, this sounds like substantial loss that should be discussed at least briefly – whether the tagging somehow affected them negatively, whether they drifted to unobserved nests and did not return, or whether this represents normal mortality.
7. The nests that were not sending or receiving drifters at all sound interesting, can any of their characteristics be identified?
8. I don’t follow the logic of interpreting the results pre and post manipulation. I appreciate

the caution in the analyses as nearly all nests are connected, but what do the results shown then? The effect of brood ratio on donors, but the boring interpretation for this is that the number of nests manipulated was small, rather than the wasps adapting to the changes in need of brood care. Also, the tables 3/4 are very difficult to interpret with the estimates in the Table 3 for brood ratios are 0.000 and -0.000, and change -0.079 and 0.138 in Table 4, i.e. switch directions? the discussion based on these results should be toned down.

9. I don't find the figures very informative. Figure 1 gives a summary of the interpretation, but nothing to justify it, and Figure just really shows that drifting is common and not only determined by distance (if the spatial layout of the colonies in the figure is based on their spatial pattern, which is not 100% clear to me). For assessing the results, it would be more informative to give some of the information now in tables as figures, but this is just a suggestion.

10. The results clearly show that the networks are more stable than the null expectation. However, it should perhaps be clearly discussed that this is not surprising as the correlates of drifting (nest size, brood ratio, location), presumably stay quite stable through the observation period.

11. Please use fewer significant figures when giving the test statistic and p-values, going down to five decimal places is not useful.

12. Remove speculative and discussive sentences from the results section, just present the results. Especially lines 369-371 - this is very speculative, what information is moved between the populations in this particular case?

Intro/discussion

13. I don't find the "multi-level social organization" framework very helpful here (esp. in the light of my comment 4), it does not really bring much to the discussion on top of what setting this study in the framework of drifting observed in other social insects would do. In the introduction drifting due to helping and drifting due to social parasitism are discussed, and I think this suffices as a framework. If a broader framework is needed, I think comparison to polydomy in ants could be more informative.

14. the reference list has some typos and strange formattings:

Ref 10 Chaline, not Cha

Ref 16: ?

Ref 33 Sokal, not Sokhal

Review form: Reviewer 3

Recommendation

Accept with minor revision (please list in comments)

Scientific importance: Is the manuscript an original and important contribution to its field?

Good

General interest: Is the paper of sufficient general interest?

Good

Quality of the paper: Is the overall quality of the paper suitable?

Good

Is the length of the paper justified?

Yes

Should the paper be seen by a specialist statistical reviewer?

No

Do you have any concerns about statistical analyses in this paper? If so, please specify them explicitly in your report.

No

It is a condition of publication that authors make their supporting data, code and materials available - either as supplementary material or hosted in an external repository. Please rate, if applicable, the supporting data on the following criteria.

Is it accessible?

N/A

Is it clear?

N/A

Is it adequate?

N/A

Do you have any ethical concerns with this paper?

No

Comments to the Author

Comments on RSPB-2020-0999

This study investigates the social structure of the paper wasp *Polistes canadensis*, using novel tracking technology and social network analyses to reveal 3 levels of social organization across which drifter wasps interact. Drifter wasps more often came from larger natal nests, moved between nests that were closer to their natal nest, and visited nests that required more help (had a higher brood to worker ratio). These results are interpreted as evidence that indirect fitness payoffs can be realised at levels beyond the nest.

I thought this manuscript was well written and enjoyable to read. A considerable amount of work has obviously gone into tagging and tracking a significant number of wasps, and experimentally manipulating demand for help, and the analysis is thorough. I do have a few, relatively minor comments on the paper, and I outline these below.

- L111: Was there a chance that tagged wasps were 'double counted' on arriving at and leaving a nest? How was this dealt with?
- L123-125: Related to the point above, this line states that analyses were performed on the number of unique drifting wasps, rather than the number of drifting events. However, at other points in the manuscript, the number of drifting events is referred to (e.g. line 174 'frequency of drifting'; L257 'number of drifting events'). This is confusing so I would suggest either using consistent language throughout (if the variable is number of unique drifters), or explaining why number of drifting wasps/ number of drifting events is analysed at each analyses.
- L126: When, and for how long, was the study period? Information on the years is given, but no further information on dates/durations.
- L148: Table 3 referenced first. Can the tables be presented in the order they are referred to in the text?
- L152: Does removing a subset of wasps change the dominance hierarchy? Could this explain changes in drifter wasp movements post-manipulation? I would welcome a bit more background ecology on the species to understand things like how frequently subordinates lay eggs (this might motivate movements between non-natal nests), whether there is a queue to inherit the nest (experimentally shortening the queue in the nest next door might motivate wasps

to visit more frequently if there is a chance they might attain dominance there more quickly, perhaps).

- L181: I'm not clear on what a binarized version of the network is. Could you briefly explain?
- Equation in L198: What does k represent?
- L228: What is the out-degree of the nest?
- L229: How many times?
- L259: It's not clear until looking at the statistical tables that nest size and worker:brood ratio of both the donor and recipient are included in models. Can you make this explicit?
- L259: How correlated are nest size and worker:brood ratio in nests? Is there a limit to the queen's egg laying capacity such that larger nests end up with a higher worker:brood ratio than smaller nests?
- L261: I'm not clear on why a logistic model (used for binary data) was used to analyse a count response variable (number of drifting events between pairs of nests). By L262-263, do you mean that you converted count data into 0/1 (drifting didn't occur/did occur) to analyse it logistically. This is not clear.
- L305-306: I'm not sure what a trace or a directed mixing matrix is. I feel this section (L302-318) could do with a little more explaining, and I'm not clear on the analyses, or what it shows.
- Methods section in general: I think it would be clearer to subhead data analyses sections using the subheadings in the Results section. In my mind, this would help link up the different analyses to what they are testing, and enable the reader to follow through to the relevant results in the next section.
- L329 & line 334: I'm not sure how or why these results are different. They read like they are the same thing (% drifters) but I'm not sure why the numbers are different.

Decision letter (RSPB-2020-0999.R0)

11-Jun-2020

Dear Dr Lengronne,

I am writing to inform you that we have now received reviews on your manuscript RSPB-2020-0999 entitled "MULTI-LEVEL SOCIAL ORGANISATION AND NEST-DRIFTING BEHAVIOUR IN A EUSOCIAL INSECT".

The manuscript has, in its current form, been rejected for publication in Proceedings B. This action has been taken on the advice of an Associate Editor and the referees, who have recommended that substantial revisions are necessary. With this in mind we would be happy to consider a resubmission, provided the comments of the referees are fully addressed. However please note that this is not a provisional acceptance.

Finally, I hope you and your co-authors are well in the current difficult times.

Yours sincerely,
 Professor Loeske Kruuk
 Editor
 mailto: proceedingsb@royalsociety.org

Associate Editor
 Board Member: 1
 Comments to Author:

Your manuscript has received reviews from three experts in the field. As you can see from their reports, they all raise issues with the manuscript. One reviewer explicitly raises the issue of whether your manuscript provides sufficiently broad and valuable results to warrant publication. I agree with this reviewer that currently you do not make this clear, but rather pitch your manuscript as of value through being comprehensive and detailed. Neither of these traits represent the kind of major step forward papers in Proceedings require. Consequently, in any revision of this manuscript (and in the response to reviewer) you need to demonstrate clearly the major steps forward in understanding from previous work in this area that this current manuscript takes, if it is to be considered for publication. Obviously, all the other issues raised also need to be convincingly addressed, both in the manuscript and the response letter.

In addition, one of the reviewers noted concern that you stated in the manuscript that "This article does not present research with ethical considerations". They noted that whilst there might not be ethical concerns, there are always ethical considerations vis a vis a) conducting fieldwork, b) carrying out work in developing countries as visiting scientists, and c) working on and killing live animals. As the Royal Society upholds the highest standards of research ethics, please reconsider your statement in this light.

Reviewer(s)' Comments to Author:
 Referee: 1

Comments to the Author(s)

This paper addresses an interesting question, namely, how is drifting behaviour between eusocial wasp nests is affected by the needs of those nests, their size and their spatial distribution. A manipulation of worker:brood ratio was used to change the need of the nests. The distance between nests and worker population were also taken into account as explanatory variables, with

distance being used as a proxy for relatedness. The methods are somewhat complex, both because of the nature of the network analysis, and because the authors have combined multiple datasets collected over several years in this paper. Despite these challenges, I view the experiment as being well-executed and the paper is very well-written. The results are interesting, not just to wasp biologists, but to those working on cooperation and social behaviour more broadly.

I have no significant concerns about the work or its presentation here. I make a few minor comments below on issues of clarity in the manuscript. These will be easy for the authors to implement, and the resulting ms will be high quality.

The methods sections contain quite a lot of results, but I don't think that is a problem – they are needed to explain the decision made in the methods, and it is clearer to include them in the methods than to make the reader track back and forth between the results and the methods to understand the rationale for the approaches taken. I am less keen on the mixing in of discussion with the results, e.g. lines 407-410 are a somewhat speculative hypothesis, and while they are consistent with the data, I don't think they merit a place in the results section. Similarly lines 420-422 of the results belong in the discussion. Lines 427-430 of the results are methods, and repeat what has already been said, so could be deleted.

When the methods are introduced, the number of wasps (presumably adults) per nest is given, but it would be helpful to the reader to state whether each nest has a single queen, or whether multiple females are reproductively active. This information is provided implicitly later, but it would be helpful to give it explicitly up front, because this aspect of ecology is variable within *Polistes*, and is highly relevant to the implications of drifting

Lines 163 and 281. I suggest a qualifier preceding 'connected', e.g. socially connected, because readers who are not familiar with paper wasp biology could assume this means structurally connected.

Line 224. I think the phrasing is a bit confusing here. It isn't the significance of the number of edges that is being determined, but whether the number of edges differs statistically significantly from the null model. Whether the number of edges is significant is a broader question (and an interesting one, but not addressed here)

Line 286. Is the correlation here also a Spearman's rho?

I think a supplementary table summarising the key differences between the datasets, and which datasets were used for which parts of the analyses would be useful – this information is given in the text, but it is hard to keep track of, and very hard to assess the implications of this, because there are so many exceptions throughout.

I wonder whether the authors still think, after their analysis, that 'drifting' is the correct term for these inter-nest movement events? Drifting rather implies an aimless unintentional behaviour, and the results don't seem consistent with this idea. I don't suggest the authors should change this throughout – that would be confusing with regard to previous literature. But perhaps they might consider raising an alternative term for this behaviour in their discussion? Or at least comment on the likely unfitness of the current term? But I leave that to the author's discretion, and realise they are likely up against the word count, given the complexity of their methods anyway.

Referee: 2

Comments to the Author(s)

This paper used RFID-tag data together with nest demography and relatedness data to describe the movement patterns of individuals in a *Polistes* wasp. As *Polistes* wasps are one of the key taxa in our current understanding of primitively eusocial insect societies, this paper has a clear

target audience in social insect researchers. The paper also introduces statistical rigor into the analyses of drifting patterns through network statistical analyses.

The main results of the paper are that, first, drifting between nests is common, second, that drifting is not random and not explained by spatial proximity alone, and third, that most movement between nests take place locally and between location drifting is rare. Furthermore, larger nests seem to send out more drifters, and also the worker to brood ratio in a nest explains some variation in drifting patterns. Given that the role of relatedness underlying drifting patterns has been described in this species earlier, and some of my concerns I detail below that undermine the robustness of some interpretations, I feel that none of these results have the broad general appeal that I would expect from a paper in this journal, and thus I see this paper more fitting in a more specialist animal behaviour or behavioural ecology journal.

Apart from the comments that follow, the data collection methods and statistical analyses are thorough, and the paper is well written and clearly argued.

Methods:

1. How was it determined which is the “home nest” of an individual from which it drifts to other colonies? I assume this is straightforward (the nest where they were originally captured?), but should be described anyway, as this is important for the speculation about indirect fitness gains: it makes a difference whether individuals from a big nest drift to help at a small nest where help is more valuable, or vice versa.
2. I’m not sure I follow the logic of lines 122-125. Why is the potential source of error smaller if only unique drifting wasps are used, can’t they be misallocated as well? Also, are these unique drifting wasps on a given observation day, or over the whole time period? If the latter, some of the results are hard to understand: lines 366-368.
3. A lot of the results are written in a way that suggests that the data allowed differentiating between a wasp entering a nest and leaving a nest (rather than just being detected at the reader), but the methods don’t give this picture, how was this done?
4. It would be nice to have some indication of how common nests of the species are outside the studied locations. If there are a lot of nests not observed, it makes a difference on the interpretation of the larger spatial scale results. Of course it is not feasible to observe every single nest within flight radius and I’m not expecting that, but this should be at least briefly discussed. This is also relevant for the speculation on the effects of anthropogenic change in the discussion.
5. the methods also contain a lot of analysis results, that should perhaps be moved to the Results section.

Results

6. Out of 1599 tagged wasps 1009 were observed again, this sounds like substantial loss that should be discussed at least briefly – whether the tagging somehow affected them negatively, whether they drifted to unobserved nests and did not return, or whether this represents normal mortality.
7. The nests that were not sending or receiving drifters at all sound interesting, can any of their characteristics be identified?
8. I don’t follow the logic of interpreting the results pre and post manipulation. I appreciate the caution in the analyses as nearly all nests are connected, but what do the results shown then? The effect of brood ratio on donors, but the boring interpretation for this is that the number of nests manipulated was small, rather than the wasps adapting to the changes in need of brood care. Also, the tables 3/4 are very difficult to interpret with the estimates in the Table 3 for brood ratios are 0.000 and -0.000, and change -0.079 and 0.138 in Table 4, i.e. switch directions? the discussion based on these results should be toned down.
9. I don’t find the figures very informative. Figure 1 gives a summary of the interpretation, but nothing to justify it, and Figure just really shows that drifting is common and not only determined by distance (if the spatial layout of the colonies in the figure is based on their spatial pattern, which is not 100% clear to me). For assessing the results, it would be more informative to give some of the information now in tables as figures, but this is just a suggestion.
10. The results clearly show that the networks are more stable than the null expectation. However, it should perhaps be clearly discussed that this is not surprising as the correlates of

drifting (nest size, brood ratio, location), presumably stay quite stable through the observation period.

11. Please use fewer significant figures when giving the test statistic and p-values, going down to five decimal places is not useful.

12. Remove speculative and discussive sentences from the results section, just present the results. Especially lines 369-371 – this is very speculative, what information is moved between the populations in this particular case?

Intro/discussion

13. I don't find the "multi-level social organization" framework very helpful here (esp. in the light of my comment 4), it does not really bring much to the discussion on top of what setting this study in the framework of drifting observed in other social insects would do. In the introduction drifting due to helping and drifting due to social parasitism are discussed, and I think this suffices as a framework. If a broader framework is needed, I think comparison to polydomy in ants could be more informative.

14. the reference list has some typos and strange formattings:

Ref 10 Chaline, not Cha

Ref 16: ?

Ref 33 Sokal, not Sokhal

Referee: 3

Comments to the Author(s)

Comments on RSPB-2020-0999

This study investigates the social structure of the paper wasp *Polistes canadensis*, using novel tracking technology and social network analyses to reveal 3 levels of social organization across which drifter wasps interact. Drifter wasps more often came from larger natal nests, moved between nests that were closer to their natal nest, and visited nests that required more help (had a higher brood to worker ratio). These results are interpreted as evidence that indirect fitness payoffs can be realised at levels beyond the nest.

I thought this manuscript was well written and enjoyable to read. A considerable amount of work has obviously gone into tagging and tracking a significant number of wasps, and experimentally manipulating demand for help, and the analysis is thorough. I do have a few, relatively minor comments on the paper, and I outline these below.

- L111: Was there a chance that tagged wasps were 'double counted' on arriving at and leaving a nest? How was this dealt with?

- L123-125: Related to the point above, this line states that analyses were performed on the number of unique drifting wasps, rather than the number of drifting events. However, at other points in the manuscript, the number of drifting events is referred to (e.g. line 174 'frequency of drifting'; L257 'number of drifting events'). This is confusing so I would suggest either using consistent language throughout (if the variable is number of unique drifters), or explaining why number of drifting wasps/number of drifting events is analysed at each analyses.

- L126: When, and for how long, was the study period? Information on the years is given, but no further information on dates/durations.

- L148: Table 3 referenced first. Can the tables be presented in the order they are referred to in the text?

- L152: Does removing a subset of wasps change the dominance hierarchy? Could this explain changes in drifter wasp movements post-manipulation? I would welcome a bit more background ecology on the species to understand things like how frequently subordinates lay eggs (this might

motivate movements between non-natal nests), whether there is a queue to inherit the nest (experimentally shortening the queue in the nest next door might motivate wasps to visit more frequently if there is a chance they might attain dominance there more quickly, perhaps).

- L181: I'm not clear on what a binarized version of the network is. Could you briefly explain?

- Equation in L198: What does k represent?

- L228: What is the out-degree of the nest?

- L229: How many times?

- L259: It's not clear until looking at the statistical tables that nest size and worker:brood ratio of both the donor and recipient are included in models. Can you make this explicit?

- L259: How correlated are nest size and worker:brood ratio in nests? Is there a limit to the queen's egg laying capacity such that larger nests end up with a higher worker:brood ratio than smaller nests?

- L261: I'm not clear on why a logistic model (used for binary data) was used to analyse a count response variable (number of drifting events between pairs of nests). By L262-263, do you mean that you converted count data into 0/1 (drifting didn't occur/did occur) to analyse it logistically. This is not clear.

- L305-306: I'm not sure what a trace or a directed mixing matrix is. I feel this section (L302-318) could do with a little more explaining, and I'm not clear on the analyses, or what it shows.

- Methods section in general: I think it would be clearer to subhead data analyses sections using the subheadings in the Results section. In my mind, this would help link up the different analyses to what they are testing, and enable the reader to follow through to the relevant results in the next section.

- L329 & line 334: I'm not sure how or why these results are different. They read like they are the same thing (% drifters) but I'm not sure why the numbers are different.

Author's Response to Decision Letter for (RSPB-2020-0999.R0)

See Appendix A.

RSPB-2020-1739.R0

Review form: Reviewer 2

Recommendation

Major revision is needed (please make suggestions in comments)

Scientific importance: Is the manuscript an original and important contribution to its field?

Good

General interest: Is the paper of sufficient general interest?

Acceptable

Quality of the paper: Is the overall quality of the paper suitable?

Good

Is the length of the paper justified?

Yes

Should the paper be seen by a specialist statistical reviewer?

Yes

Do you have any concerns about statistical analyses in this paper? If so, please specify them explicitly in your report.

Yes

It is a condition of publication that authors make their supporting data, code and materials available - either as supplementary material or hosted in an external repository. Please rate, if applicable, the supporting data on the following criteria.

Is it accessible?

Yes

Is it clear?

Yes

Is it adequate?

Yes

Do you have any ethical concerns with this paper?

No

Comments to the Author

This is a revision of a paper I have assessed in the first round as well. I have to say I still struggle with its value, as the main important outcome (the adaptive significance of drifting shown by experimental manipulations) is still quite weak in its interpretation. Being comprehensive and rigorous, as well as having data that a lot of effort went into, are not to me enough for a high profile journal if the results do not provide interesting new insight, robustly supported. I details my thoughts on the main results below.

1. To me the non-randomness of drifting is not really a novel result, as the earlier demonstration that drifting correlates with relatedness already shows non-randomness, and suggests adaptive value. Here it is shown that spatial distance explains drifting (but not simply as a function of diffusion), and is highly correlated with relatedness, and it remains unclear what is a cause and what is a consequence here, as due to lacking relatedness data spatial distance only is used. In this light, it is the variables explaining drifting patterns that are of the highest interest, and their discussion need strengthening.
2. The finding that more individuals drift from larger colonies is not surprising as such, as that does not demonstrate that an individual worker is more likely to drift from a large colony (i.e. this result could be an outcome of a stable proportion of workers always drifting, which sounds less like adaptive adjustment of worker behaviour). Perhaps more could be understood if the proportions of workers drifting were analysed, if the claim is that workers adjust their behaviour to local conditions? This might strengthen the reasoning that workers make adaptive decisions.
3. The main novel result is that (pre-manipulation), worker-brood ratio of both the donor and recipient explain variation in drifting. Here, however, the effect sizes are tiny, even if significant. The authors now state in the Table legends this is because of low statistical power, but isn't it the

opposite: a tiny effect size, highly statistically significant sounds like a very powerful analysis, difficult to interpret biologically? Without expertise in these particular methods (the different QAP variants), it is very difficult to interpret the findings, especially since w/b ratios also highly significantly correlate with colony size. I strongly recommend additional analyses to support the conclusions here, even if I unfortunately can't offer advice on how this could be done – just seeing that the same results could be retrieved with a traditional binomial GLMM would be reassuring, even if the data points are not truly independent. Visual presentation of the results could also be helpful, but I guess such tiny effect sizes suggest that not much would be seen in scatterplots or plots of model predictions or similar.

4. The post-manipulation results repeat some, but not all of the pre-manipulation results: donor nest size and recipient nest w/b ratio lose significance (although effect sizes interestingly stay high, which also is a bit strange!). The lack of a nest size response, but the maintenance of a worker-brood ratio effect could be interpreted as workers making decisions based on the local needs. However, they don't seem to be able to direct their efforts into recipient nests according to their needs, which limits the adaptive value. This should be discussed.

5. the “multi-level” framework still feels unnecessary to me, it would suffice to say that “drifting is very rare between aggregates”, and this would leave more space for discussing the possible adaptive features within aggregates.

I have also have some smaller comments on the responses to my earlier comments:

My comment: 2.4. It would be nice to have some indication of how common nests of the species are outside the studied locations. If there are a lot of nests not observed, it makes a difference on the interpretation of the larger spatial scale results. Of course it is not feasible to observe every single nest within flight radius and I'm not expecting that, but this should be at least briefly discussed. This is also relevant for the speculation on the effects of anthropogenic change in the discussion.

Response: 2.4 We added a test to indicate that nests are rarely found alone; they tend to be spatially discrete, in aggregations. This applies to populations in natural substrates (e.g., trees, caves) as well as anthropogenic substrates like buildings.

My response: This partly answers my concern. But the main question really is then whether it is likely that there are aggregations of nests in the vicinity, or between the sampled aggregates, that were not sampled? Is it possible that there are missed aggregations where to more drifting occurs?

My comment 2.7. The nests that were not sending or receiving drifters at all sound interesting, can any of their characteristics be identified?

Response 2.7 Although we did not address this explicitly, the answer is in fact in the reciprocal interpretation of the data presented: drifters are more likely to be sent from large nests and are more likely to be received by small nests.

My response: this is confusing: by this logic a nest that neither sends nor receives drifters is a) small because it does not send drifters b) not small because it does not receive drifters. But maybe in the interest of space limits this is not important, if these nests did not have any particular characteristics.

Figure 1: one suggestion towards how this could be made a bit more informative is that the location of the nests would be presented according to their spatial lay-out – at least then the effects of spatial proximity would be shown, although perhaps not in a very clear way. Still, this Figure tells very little of the results.

Decision letter (RSPB-2020-1739.R0)

13-Aug-2020

Dear Dr Lengronne,

I am writing to inform you that we have now received reviewers' reports on your revised version of your manuscript RSPB-2020-1739 entitled "MULTI-LEVEL SOCIAL ORGANISATION AND NEST-DRIFTING BEHAVIOUR IN A EUSOCIAL INSECT". This was reviewed again in detail by one of the original reviewers and the Associate Editor.

We appreciate the work that you have put into the revised version, but I regret to say that there are still several substantial concerns about the paper. These are set out in the AE's and reviewer's reports below, and indicate that substantial revisions are necessary. The paper has therefore been rejected in its current form. However we would be willing to consider a resubmission, provided the issues that have been raised are fully addressed, including re-analysis of the data following the suggestions below. However please note that this is not a provisional acceptance.

- 1) A 'response to referees' document including details of how you have responded to the comments, and the adjustments you have made.
- 2) A clean copy of the manuscript and one with 'tracked changes' indicating your 'response to referees' comments document.
- 3) Line numbers in your main document.
- 4) Please read our data sharing policies to ensure that you meet our requirements <https://royalsociety.org/journals/authors/author-guidelines/#data>.

Finally, I hope you and your co-authors are well in these strange times.

Yours sincerely,
Professor Loeske Kruuk
Editor
<mailto:proceedingsb@royalsociety.org>

Associate Editor
Comments to Author:

Thank you for your engagement with the reviewer comments, and the subsequent response letter and revisions on the manuscript. These have been seen by one of the reviewers again, who has provided a second round of review. As you can see, this still raises significant issues around the novelty and broader value of the manuscript. In particular, the following points concern me:

1) that the non-randomness of drifting is novel, when the senior author and colleagues already showed this (vis-a-vis going to genetically related nests), and the metric you use here, distance, you show to correlate with relationship. What is the big step forward here?

2) that your analyses of the impact of nest size on drifting look at numbers of events. It's not clear to me what your null hypothesis is here. If any given wasp has a probability of drifting (a reasonable null hypothesis?), then simply by virtue of nest size, larger nests will produce more drifters, and vice versa for smaller nests. This is not particularly biologically interesting. You at least need to account for this, using some kind of proportional measure in your analyses, to show whether the patterns you see are simply a function of such a null model, or whether large nests produce a disproportionately higher number of drifting events than would be expected based on just size. If you have already done this, it's not clear from your methods and results section.

3) that your analyses of other drivers of drifting show inconsistent results, with very small effect sizes. It is really hard to know how much weight to put on these in terms of biological meaningfulness (also, as per the reviewer, I don't follow your argument about power and effect sizes and significance)

4) the multi-level social organisation approach is interesting, but is not entirely novel (even to your own system). The senior author already showed the first two components of it in the *Current Biology* paper that, presumably, inspired the current study. That wasps move between aggregations is, perhaps, surprising, but hard to interpret as your own results show that drifting is largely driven by the relatedness/distance effect, and how cross aggregation drifting relates to this driver is unclear in the absence of relatedness data for these drifters and the nests they went to, and given that how the cross-aggregation distance relates to the scale at which you have previously identified distance/relatedness as being important is unclear. In addition, the 'anthropocene' spin you put on this, in the absence of any data on wasps in natural settings, is pure speculation

5) that you haven't addressed some of the points raised previously by this reviewer (which they have raised again)

I do believe that this is a valuable manuscript, but it needs to be carefully revised bearing the points above, and the more detailed points made by the reviewer, in mind. In particular, I believe you need to re-analyse your data for the nest size effect. Finally, please make crystal clear in the revised document exactly what the steps forward are, and why they matter. There is still a greater emphasis on the size and scale of the study, as opposed to what we are learning from it that makes it of broad interest to scientists.

Reviewer(s)' Comments to Author:

Referee: 2

Comments to the Author(s).

This is a revision of a paper I have assessed in the first round as well. I have to say I still struggle with its value, as the main important outcome (the adaptive significance of drifting shown by experimental manipulations) is still quite weak in its interpretation. Being comprehensive and rigorous, as well as having data that a lot of effort went into, are not to me enough for a high profile journal if the results do not provide interesting new insight, robustly supported. I detail my thoughts on the main results below.

1. To me the non-randomness of drifting is not really a novel result, as the earlier demonstration that drifting correlates with relatedness already shows non-randomness, and suggests adaptive value. Here it is shown that spatial distance explains drifting (but not simply as a function of diffusion), and is highly correlated with relatedness, and it remains unclear what is a cause and what is a consequence here, as due to lacking relatedness data spatial distance only is used. In

this light, it is the variables explaining drifting patterns that are of the highest interest, and their discussion need strengthening.

2. The finding that more individuals drift from larger colonies is not surprising as such, as that does not demonstrate that an individual worker is more likely to drift from a large colony (i.e. this result could be an outcome of a stable proportion of workers always drifting, which sounds less like adaptive adjustment of worker behaviour). Perhaps more could be understood if the proportions of workers drifting were analysed, if the claim is that workers adjust their behaviour to local conditions? This might strengthen the reasoning that workers make adaptive decisions.
3. The main novel result is that (pre-manipulation), worker-brood ratio of both the donor and recipient explain variation in drifting. Here, however, the effect sizes are tiny, even if significant. The authors now state in the Table legends this is because of low statistical power, but isn't it the opposite: a tiny effect size, highly statistically significant sounds like a very powerful analysis, difficult to interpret biologically? Without expertise in these particular methods (the different QAP variants), it is very difficult to interpret the findings, especially since w/b ratios also highly significantly correlate with colony size. I strongly recommend additional analyses to support the conclusions here, even if I unfortunately can't offer advice on how this could be done – just seeing that the same results could be retrieved with a traditional binomial GLMM would be reassuring, even if the data points are not truly independent. Visual presentation of the results could also be helpful, but I guess such tiny effect sizes suggest that not much would be seen in scatterplots or plots of model predictions or similar.
4. The post-manipulation results repeat some, but not all of the pre-manipulation results: donor nest size and recipient nest w/b ratio lose significance (although effect sizes interestingly stay high, which also is a bit strange!). The lack of a nest size response, but the maintenance of a worker-brood ratio effect could be interpreted as workers making decisions based on the local needs. However, they don't seem to be able to direct their efforts into recipient nests according to their needs, which limits the adaptive value. This should be discussed.
5. the “multi-level” framework still feels unnecessary to me, it would suffice to say that “drifting is very rare between aggregates”, and this would leave more space for discussing the possible adaptive features within aggregates.

I have also have some smaller comments on the responses to my earlier comments:

My comment: 2.4. It would be nice to have some indication of how common nests of the species are outside the studied locations. If there are a lot of nests not observed, it makes a difference on the interpretation of the larger spatial scale results. Of course it is not feasible to observe every single nest within flight radius and I'm not expecting that, but this should be at least briefly discussed. This is also relevant for the speculation on the effects of anthropogenic change in the discussion.

Response: 2.4 We added a test to indicate that nests are rarely found alone; they tend to be spatially discrete, in aggregations. This applies to populations in natural substrates (e.g., trees, caves) as well as anthropogenic substrates like buildings.

My response: This partly answers my concern. But the main question really is then whether it is likely that there are aggregations of nests in the vicinity, or between the sampled aggregates, that were not sampled? Is it possible that there are missed aggregations where to more drifting occurs?

My comment 2.7. The nests that were not sending or receiving drifters at all sound interesting, can any of their characteristics be identified?

Response 2.7 Although we did not address this explicitly, the answer is in fact in the reciprocal interpretation of the data presented: drifters are more likely to be sent from large nests and are more likely to be received by small nests.

My response: this is confusing: by this logic a nest that neither sends nor receives drifters is a) small because it does not send drifters b) not small because it does not receive drifters. But maybe in the interest of space limits this is not important, if these nests did not have any particular characteristics.

Figure 1: one suggestion towards how this could be made a bit more informative is that the location of the nests would be presented according to their spatial lay-out – at least then the effects of spatial proximity would be shown, although perhaps not in a very clear way. Still, this Figure tells very little of the results.

Author's Response to Decision Letter for (RSPB-2020-1739.R0)

See Appendix B.

RSPB-2021-0275.R0

Review form: Reviewer 2

Recommendation

Reject – article is not of sufficient interest (we will consider a transfer to another journal)

Scientific importance: Is the manuscript an original and important contribution to its field?

Good

General interest: Is the paper of sufficient general interest?

Acceptable

Quality of the paper: Is the overall quality of the paper suitable?

Good

Is the length of the paper justified?

Yes

Should the paper be seen by a specialist statistical reviewer?

No

Do you have any concerns about statistical analyses in this paper? If so, please specify them explicitly in your report.

No

It is a condition of publication that authors make their supporting data, code and materials available - either as supplementary material or hosted in an external repository. Please rate, if applicable, the supporting data on the following criteria.

Is it accessible?

Yes

Is it clear?

Yes

Is it adequate?

Yes

Do you have any ethical concerns with this paper?

No

Comments to the Author

This is a re-revision of a paper I have assessed previously. The authors have put a lot of effort into explaining the importance of the paper – especially the rigorous methods not used for this kind of social insect data previously, and the sheer size of the data. I agree that the methodological rigor is welcome, but that would be much stronger in a paper focusing more on the methods, spending more time to explain the problems, and the value of the current analyses. It remained unclear to me what exactly the main reason for the need of the novel methods is: the lack of independence of observations from one nest to multiple other nests, so that each pairwise measure is not independent (much like when a Mantel test is used in significance testing of pairwise distance matrices), or the correlations or collinearities between explanatory variables (which is a problem not limited to network data), or both, or something else. The value of the paper in this direction could be much improved, but in this case I think some more specialized journal would be suitable. The deeper focus on the methods could also help explain the puzzling finding of high significance levels for tiny effect sizes.

But my main criticism still stands: “the main important outcome (the adaptive significance of drifting shown by experimental manipulations) is still quite weak in its interpretation.” The authors have not made changes in this respect, and in their discussion put a lot of emphasis on this weaker aspect of the paper, rather than the strengths.

In more detail:

It is shown that the number of individuals to leave the nest is affected by local features of the donor nest: pre-manipulation donor size, pre-manipulation donor w/b-ratio (with an effect size that is very difficult to see as biologically meaningful), and post manipulation w/b-ratio. The effect of donor size could be simply due to more workers being there – the proportional chances of a single worker leaving the nest were not analysed, even though I suggested it could be helpful, and AE seems to have been worried about the same thing. I don't understand why these analyses were not carried out – they could have helped make more sense of the data. The issue that I recommended discussing, that there is very little support for drifting TO the nests with the high needs, is not properly discussed.

The only significant feature of the recipient nest is the pre-manipulation w/b ratio, with an effect size that is tiny, and unlikely to be biologically significant. Despite this, the authors claim in the discussion

510-512: “By drifting to smaller nests, nests which are close by (and thus tend to be more closely related), and by drifting from nests that have lots of workers relative to the brood number”

535-538: “That larger nests donate more drifters, and smaller nests receive more drifters makes sense since the added value of each worker diminishes with increasing group size in many eusocial insects”

These claims are not supported but the data.

538-541 “By giving help preferentially to groups where the need for help and indirect fitness payoffs are greatest, the fitness implications of interactions at levels above and beyond the family group (nest) are likely to be substantial and should be taken into consideration when attempting to quantify”

The only evidence for this I see in the pre-manipulation (very) small effect of w/b ratio, so “likely to be substantial” seems out of proportion. A much more cautious discussion would have been in place.

Minor comments:

- the recent paper by Kennedy, Sumner et al in Nature Ecol Evol should be taken into account in the introduction and discussion

-I appreciate the addition of the GLM-results at the end, despite the associated problems, but I don't understand why a GLM-was given in one case and a ChiSquare test in the other – why not a GLM also in the second case? Especially when this Chi-Sq test is based on comparing pre/post manipulation patterns – something the themselves authors claim is unjustified.

Decision letter (RSPB-2021-0275.R0)

24-Mar-2021

Dear Dr Sumner

I am pleased to inform you that your manuscript RSPB-2021-0275 entitled "MULTI-LEVEL SOCIAL ORGANISATION AND NEST-DRIFTING BEHAVIOUR IN A EUSOCIAL INSECT" has been accepted for publication in Proceedings B.

Thank you for the extensive revisions and response to comments. The revised version of your paper was reviewed by one of the original referees, and you will see from their comments that they still have problems with several aspects of the paper. However the Associate Editor and I are happy that you have made the effort to address the earlier main reservations of the paper, and hence to accept this with minor revisions: for the final version I ask that you do what is feasible to address the referee's points in their review below, include reference to your other recent relevant publication, and address my comments below regarding the GLMMs. Because the schedule for publication is very tight, it is a condition of publication that you submit the revised version of your manuscript within 7 days. If you do not think you will be able to meet this date please let us know.

Online supplementary material will also carry the title and description provided during submission, so please ensure these are accurate and informative. Note that the Royal Society will not edit or typeset supplementary material and it will be hosted as provided. Please ensure that

the supplementary material includes the paper details (authors, title, journal name, article DOI). Your article DOI will be 10.1098/rspb.[paper ID in form xxxx.xxxx e.g. 10.1098/rspb.2016.0049].

Once again, thank you for submitting your manuscript to Proceedings B; I realise the manuscript has had a long journey, and I am looking forward to receiving the final revision. If you have any questions at all, please do not hesitate to get in touch.

Yours sincerely,
 Professor Loeske Kruuk
 Editor
 mailto: proceedingsb@royalsociety.org

Associate Editor
 Board Member
 Comments to Author:

I have gone through your revised manuscript, the most recent review, and the history of reviews. In my opinion your manuscript has significantly improved over the review process, and its novelty and value are now clear. During this process, however, a subset of the authors have published a related paper (Kennedy et al 2021 NEE) which needs to be integrated and cited in your Introduction and Discussion sections of the current manuscript.

Editor comments (LK)

Thank you for including the GLMM information in the Suppl Info, as per the reviewer's original request, but please also provide standard full information on these models (e.g. what fixed and random effects were specified, parameter estimates, variance components, etc.).

Regarding your statement that these models are not appropriate, as per previous comments, GLMMs are required to deal with structure and hence non-independence of response variables. Correlations between the explanatory variables do not render them inappropriate, as currently claimed in the SI: see for example Morrissey, M. B. and G. D. Ruxton. 2018. "Multiple Regression Is Not Multiple Regressions: The Meaning of Multiple Regression and the Non-Problem of Collinearity." *Philosophy, Theory, and Practice in Biology* 10, which addresses misconceptions about collinearity. Where explanatory variables are correlated, care needs to be taken in their interpretation (hence needing to see the full output of the parameter estimates), but this does not render the models invalid.

The Supplementary Information needs a proof-read (e.g. remove comments in margins, typo in final sentence of penultimate paragraph, be consistent with talking about either in-degree or indegree). And it refers to a table ("Table below"), but does not include one; tables are needed for the output of each of the models referred to.

Reviewer(s)' Comments to Author:

Referee: 2

Comments to the Author(s).

This is a re-revision of a paper I have assessed previously. The authors have put a lot of effort into explaining the importance of the paper – especially the rigorous methods not used for this kind of social insect data previously, and the sheer size of the data. I agree that the methodological rigor is welcome, but that would be much stronger in a paper focusing more on the methods, spending more time to explain the problems, and the value of the current analyses. It remained unclear to me what exactly the main reason for the need of the novel methods is: the lack of independence of observations from one nest to multiple other nests, so that each pairwise measure is not independent (much like when a Mantel test is used in significance testing of pairwise distance matrices), or the correlations or collinearities between explanatory variables (which is a problem not limited to network data), or both, or something else. The value of the paper in this direction could be much improved, but in this case I think some more specialized journal would be suitable. The deeper focus on the methods could also help explain the puzzling finding of high significance levels for tiny effect sizes.

But my main criticism still stands: “the main important outcome (the adaptive significance of drifting shown by experimental manipulations) is still quite weak in its interpretation.” The authors have not made changes in this respect, and in their discussion put a lot of emphasis on this weaker aspect of the paper, rather than the strengths.

In more detail:

It is shown that the number of individuals to leave the nest is affected by local features of the donor nest: pre-manipulation donor size, pre-manipulation donor w/b-ratio (with an effect size that is very difficult to see as biologically meaningful), and post manipulation w/b-ratio. The effect of donor size could be simply due to more workers being there – the proportional chances of a single worker leaving the nest were not analysed, even though I suggested it could be helpful, and AE seems to have been worried about the same thing. I don't understand why these analyses were not carried out - they could have helped make more sense of the data. The issue that I recommended discussing, that there is very little support for drifting TO the nests with the high needs, is not properly discussed.

The only significant feature of the recipient nest is the pre-manipulation w/b ratio, with an effect size that is tiny, and unlikely to be biologically significant. Despite this, the authors claim in the discussion

510-512: "By drifting to smaller nests, nests which are close by (and thus tend to be more closely related), and by drifting from nests that have lots of workers relative to the brood number"

535-538: "That larger nests donate more drifters, and smaller nests receive more drifters makes sense since the added value of each worker diminishes with increasing group size in many eusocial insects"

These claims are not supported but the data.

538-541 "By giving help preferentially to groups where the need for help and indirect fitness payoffs are greatest, the fitness implications of interactions at levels above and beyond the family group (nest) are likely to be substantial and should be taken into consideration when attempting to quantify"

The only evidence for this I see in the pre-manipulation (very) small effect of w/b ratio, so "likely to be substantial" seems out of proportion. A much more cautious discussion would have been in place.

Minor comments:

- the recent paper by Kennedy, Sumner et al in Nature Ecol Evol should be taken into account in the introduction and discussion

-I appreciate the addition of the GLM-results at the end, despite the associated problems, but I don't understand why a GLM-was given in one case and a ChiSquare test in the other - why not a GLM also in the second case? Especially when this Chi-Sq test is based on comparing pre/post manipulation patterns - something the themselves authors claim is unjustified.

Author's Response to Decision Letter for (RSPB-2021-0275.R0)

See Appendix C.

Decision letter (RSPB-2021-0275.R1)

01-Apr-2021

Dear Dr Sumner

I am pleased to inform you that your manuscript entitled "MULTI-LEVEL SOCIAL ORGANISATION AND NEST-DRIFTING BEHAVIOUR IN A EUSOCIAL INSECT" has been accepted for publication in Proceedings B.

Data Accessibility section

Open Access

Paper charges

Sincerely,

Proceedings B

Appendix A

Resubmission of MS ID RSPB-2020-0999

Langronne et al: MULTI-LEVEL SOCIAL ORGANISATION AND NEST-DRIFTING BEHAVIOUR IN A EUSOCIAL INSECT

Comments to Author:

Your manuscript has received reviews from three experts in the field. As you can see from their reports, they all raise issues with the manuscript. One reviewer explicitly raises the issue of whether your manuscript provides sufficiently broad and valuable results to warrant publication. I agree with this reviewer that currently you do not make this clear, but rather pitch your manuscript as of value through being comprehensive and detailed. Neither of these traits represent the kind of major step forward papers in Proceedings require.

Consequently, in any revision of this manuscript (and in the response to reviewer) you need to demonstrate clearly the major steps forward in understanding from previous work in this area that this current manuscript takes, if it is to be considered for publication. Obviously, all the other issues raised also need to be convincingly addressed, both in the manuscript and the response letter.

We'd like to thank the reviewers for their very careful comments; they have helped improve the clarity of the MS enormously. We detail our response to all their comments below, and indicate changes made in the revised MS using numbered responses, highlighted using comments (e.g. Rev 1.2) in the revised document rather than line numbers which are easily confused.

We are surprised by yours and one reviewer's comment that our previous version did not properly make clear the ways in which our work presents a major step forward in our understanding of this unusual behaviour. Indeed, our study is one of a small handful of papers on drifting behaviours in social insects; as well as being the most comprehensive and analytically rigorous to date, it also presents novel and important insights into the adaptive nature of drifting behaviour, which is likely to be of general importance to our understanding of social evolution. We now make this clear in our response below and the revised MS.

In addition, one of the reviewers noted concern that you stated in the manuscript that "This article does not present research with ethical considerations". They noted that whilst there might not be ethical concerns, there are always ethical considerations vis a vis a) conducting fieldwork, b) carrying out work in developing countries as visiting scientists, and c) working on and killing live animals. As the Royal Society upholds the highest standards of research ethics, please reconsider your statement in this light.

Apologies, this must have been ticked in error – all work conducted in this study was with ethical consideration.

Reviewer(s)' Comments to Author:

Referee: 1

Comments to the Author(s)

This paper addresses an interesting question, namely, how is drifting behaviour between eusocial wasp nests is affected by the needs of those nests, their size and their spatial distribution. A manipulation of worker:brood ratio was used to change the need of the nests. The distance between nests and worker population were also taken into account as explanatory variables, with distance being used as a proxy for relatedness. The methods are somewhat complex, both because of the nature of the network analysis, and because the authors have combined multiple datasets collected over several years in this paper. Despite these challenges, I view the experiment as being well-executed and the paper is very well-written. The results are interesting, not just to wasp biologists, but to those working on cooperation and social behaviour more broadly.

We are delighted that the referee appreciates the volume of work in our study, found the work interesting and of general interest for researchers working on cooperation and social behaviour.

I have no significant concerns about the work or its presentation here. I make a few minor comments below on issues of clarity in the manuscript. These will be easy for the authors to implement, and the resulting ms will be high quality.

1.1 The methods sections contain quite a lot of results, but I don't think that is a problem – they are needed to explain the decision made in the methods, and it is clearer to include them in the methods than to make the reader track back and forth between the results and the methods to understand the rationale for the approaches taken.

1.1 We have thought carefully about whether any of these parts of the methods could be logically removed to the results; we agree with the reviewer that they are in fact integral to the logical flow of the methods section.

1.2 I am less keen on the mixing in of discussion with the results, e.g. lines 407-410 are a somewhat speculative hypothesis, and while they are consistent with the data, I don't think they merit a place in the results section. Similarly lines 420-422 of the results belong in the discussion. Lines 427-430 of the results are methods, and repeat what has already been said, so could be deleted.

1.2 We agree with the reviewer's comment. We have deleted the sentences on lines 407-410, 420-422 and 427-430 and reworded the relevant sections.

1.3 When the methods are introduced, the number of wasps (presumably adults) per nest is given, but it would be helpful to the reader to state whether each nest has a single queen, or whether multiple females are reproductively active. This information is provided implicitly later, but it would be helpful to give it explicitly up front, because this aspect of ecology is variable within *Polistes*, and is highly relevant to the implications of drifting

1.3 This is a very good point, and especially now that the part of the text referred to has moved to the Results (as requested in comment 1.2). We now add this into the Introduction.

1.4 Lines 163 and 281. I suggest a qualifier preceding 'connected', e.g. socially connected, because readers who are not familiar with paper wasp biology could assume this means structurally connected.

1.4 We agree and have added 'socially' as suggested.

1.5 Line 224. I think the phrasing is a bit confusing here. It isn't the significance of the number of edges that is being determined, but whether the number of edges differs statistically significantly from the null model. Whether the number of edges is significant is a broader question (and an interesting one, but not addressed here)

1.5 We agree. We have reworded the sentence accordingly.

Line 286. Is the correlation here also a Spearman's rho?

This has now been removed.

1.6 I think a supplementary table summarising the key differences between the datasets, and which datasets were used for which parts of the analyses would be useful – this information is given in the text, but it is hard to keep track of, and very hard to assess the implications of this, because there are so many exceptions throughout.

1.6 We now provide this as requested. (Suppl Table A1).

1.7 I wonder whether the authors still think, after their analysis, that ‘drifting’ is the correct term for these inter-nest movement events? Drifting rather implies an aimless unintentional behaviour, and the results don’t seem consistent with this idea. I don’t suggest the authors should change this throughout – that would be confusing with regard to previous literature. But perhaps they might consider raising an alternative term for this behaviour in their discussion? Or at least comment on the likely unfitness of the current term? But I leave that to the author’s discretion, and realise they are likely up against the word count, given the complexity of their methods anyway.

1.7 This is a very good point. We have added two sentences in the discussion to address this point.

Referee: 2

Comments to the Author(s)

2.1 This paper used RFID-tag data together with nest demography and relatedness data to describe the movement patterns of individuals in a *Polistes* wasp. As *Polistes* wasps are one of the key taxa in our current understanding of primitively eusocial insect societies, this paper has a clear target audience in social insect researchers. The paper also introduces statistical rigor into the analyses of drifting patterns through network statistical analyses. The main results of the paper are that, first, drifting between nests is common, second, that drifting is not random and not explained by spatial proximity alone, and third, that most movement between nests take place locally and between location drifting is rare. Furthermore, larger nests seem to send out more drifters, and also the worker to brood ratio in a nest explains some variation in drifting patterns. Given that the role of relatedness underlying drifting patterns has been described in this species earlier, and some of my concerns I detail below that undermine the robustness of some interpretations, I feel that none of these results have the broad general appeal that I would expect from a paper in this journal, and thus I see this paper more fitting in a more specialist animal behaviour or behavioural ecology journal.

The reviewer is correct that the role of relatedness had been previously demonstrated. However, this is the first study that attempts to experimentally manipulate important parameters that may affect the cost/benefit of drifting. Also, we demonstrate for the first time the role of nest size, nest proximity and worker:brood ratio as factors influencing the rate of drifting. This is thus the first study to demonstrate that drifting might have some adaptive value. We have now extended the discussion to discuss this as suggested by Reviewer 1. We hope that these explanations together with changes will convince the reviewer that this paper should be of general interest for readers interested in social behaviour.

2.2 Apart from the comments that follow, the data collection methods and statistical analyses are thorough, and the paper is well written and clearly argued.

Methods:

1. How was it determined which is the “home nest” of an individual from which it drifts to other colonies? I assume this is straightforward (the nest where they were originally captured?), but should be described anyway, as this is important for the speculation about indirect fitness gains: it makes a difference whether individuals from a big nest drift to help at a small nest where help is more valuable, or vice versa.

2.1 This is a good point. We add text to explain this.

2.2 I'm not sure I follow the logic of lines 122-125. Why is the potential source of error smaller if only unique drifting wasps are used, can't they be misallocated as well? Also, are these unique drifting wasps on a given observation day, or over the whole time period? If the latter, some of the results are hard to understand: lines 366-368.

2.2 Apologies, this sentence was misleading and has been removed. Analyses are of unique wasps over a specified monitoring period but also their visit numbers, depending on the analyses; this is indicated in the relevant analytical section (e.g., lines 218-222 use unique wasp identities because we were interested in determining which nests were socially connected; conversely, see lines 289 where we used number of drifting events, as we were interested in investment by drifters (as a measure of foraging effort) in different nests).

2.3. A lot of the results are written in a way that suggests that the data allowed differentiating between a wasp entering a nest and leaving a nest (rather than just being detected at the reader), but the methods don't give this picture, how was this done?

2.3 This also relates to comment 2.1 (i.e., defining the 'home (or 'donor') nest'); we have added text to explain this.

2.4. It would be nice to have some indication of how common nests of the species are outside the studied locations. If there are a lot of nests not observed, it makes a difference on the interpretation of the larger spatial scale results. Of course it is not feasible to observe every single nest within flight radius and I'm not expecting that, but this should be at least briefly discussed. This is also relevant for the speculation on the effects of anthropogenic change in the discussion.

2.4 We added a test to indicate that nests are rarely found alone; they tend to be spatially discrete, in aggregations. This applies to populations in natural substrates (e.g., trees, caves) as well as anthropogenic substrates like buildings.

2.5. the methods also contain a lot of analysis results, that should perhaps be moved to the Results section.

2.5 We agree with the reviewer. We have moved parts of the first section of the Methods section which described the details on the nests and sites analysed to the result section.

Results

2.6. Out of 1599 tagged wasps 1009 were observed again, this sounds like substantial loss that should be discussed at least briefly – whether the tagging somehow affected them negatively, whether they drifted to unobserved nests and did not return, or whether this represents normal mortality.

2.6 This re-detection rate (63%) is actually better than previous RFID studies on this species (e.g., Sumner et al. 2007 detected 157/422 (37%) tagged wasps), possibly because our tagging methods have improved (we place the tagged wasps directly back onto the nest; we do not tag newly emerged wasps until they are >3 days old). But, nonetheless, there clearly is a high disappearance rate in these wasps ('mortality' rate per wasp per day was 7% in Sumner et al 2007) and this equally applies to paint-marked and RFID tagged wasps. We also see a high turnover in the nestmate identity from other studies; we are confident that the high turnover is natural, and not caused by us, because the colony genetic structure of these nests is consistent for completely untouched nests and nests we have studied (see Southon et al. 2019). We are also confident that these wasps are not usually relocating to other nests as a result of our intervention, as in all our studies we usually monitor all nests in the vicinity,

including adjacent aggregations (as in this study), and (as reported in this study) movement between aggregations is rare). Moreover, Sumner et al. 2007 showed that tagging per se was not causing wasp to drift, as relatedness of untagged and tagged wasps was no different. It is impossible to say whether the high disappearance rate of these wasps is due to disturbance by the observers per se, as by definition we cannot know if wasps have left, unless we have marked them. This is a lot of detail to add into the current over-long MS and so we would prefer not adding more here. We hope this satisfies the reviewer.

2.7. The nests that were not sending or receiving drifters at all sound interesting, can any of their characteristics be identified?

2.7 Although we did not address this explicitly, the answer is in fact in the reciprocal interpretation of the data presented: drifters are more likely to be sent from large nests and are more likely to be received by small nests.

2.8 I don't follow the logic of interpreting the results pre and post manipulation. I appreciate the caution in the analyses as nearly all nests are connected, but what do the results show then? The effect of brood ratio on donors, but the boring interpretation for this is that the number of nests manipulated was small, rather than the wasps adapting to the changes in need of brood care. Also, the tables 3/4 are very difficult to interpret with the estimates in the Table 3 for brood ratios are 0.000 and -0.000, and change -0.079 and 0.138 in Table 4, i.e. switch directions? the discussion based on these results should be toned down.

2.8 Apologies that this was not clear. Our original experimental plan was to compare pre- and post-manipulation drifting rates and thus test the hypothesis that alteration of worker:brood ratios (i.e., increase or decrease need for help) would instigate reciprocal changes in drifting. However, this experiment was designed (and executed) without a true understanding of the high degree of connectedness that these wasp aggregations display: indeed, this is exactly one of the main interesting findings of our study. In retrospect, manipulating a single pair of nests per aggregation would be the safest option, but this would generate very small sample sizes without a herculean effort in tagging many more aggregations at enormous expense. As such, the analyses of our manipulation experiment are not as straight forward as we had hoped. The most conservative approach therefore was to analyse the pre-manipulation data and the post-manipulation data as separate ('independent') tests against the same null model of drifting. We improve our explanation of this in lines 188-193.

How to interpret the data. Each dataset is tested against a null model for whether there are significant higher or lower levels of drifting. The pre-manipulation results (Table 3) clearly show *higher* levels of drifting as distance between nests *decreases* ($p < 0.002$; estimate = -0.003), and *lower* levels of drifting as donor nest size (i.e., the nests sending drifters) *increases* ($p = 0.0012$; estimate = 0.02) (highlighted in bold now for clarity). These patterns are in line with drifting being in the direction of providing help where most needed (i.e., nests with more brood per worker). In the pre-manipulation experiment, the p values also suggest a significant effect of worker:brood ratios on drifting; however, the estimates are tiny (0.000, to 3 decimal places) (due to lack of power, as the reviewer points out) and so this result should be taken with caution. We add this into the legend for Table 3 now for clarity.

Table 4 presents exactly the same analysis but for nests in the aggregations that we manipulated. Here, the effects of distance on drifting are significant, as in pre-manipulation ($p = 0.019$; estimate = -0.002), and the same trends in nest size and drifting are apparent but not significant (and thus not highlighted in bold). But, there is less drifting as worker:brood ratios decrease ($p = 0.006$; estimate = -0.079), in line with our predictions that wasps should drift less to nests less in need of help.

Yes, the apparent direction of the estimates are opposite in Tables 3 & 4, but since those in Table 3 are not significant (and indeed extremely small) there can be little confidence placed in this comparison. However, the reviewer is right that our sample sizes are small and our effect sizes reflect this: in our defence, this work is incredibly labour intensive and tricky to do on wild populations of wasps, especially with high wasp turnover and mortality rates (noted above). We now add a comment on this in the legend for Table 3 and have also toned down our interpretation of these experiments, and added extra cautionary notes.

2.9 I don't find the figures very informative. Figure 1 gives a summary of the interpretation, but nothing to justify it, and Figure just really shows that drifting is common and not only determined by distance (if the spatial layout of the colonies in the figure is based on their spatial pattern, which is not 100% clear to me). For assessing the results, it would be more informative to give some of the information now in tables as figures, but this is just a suggestion.

2.9 We have thought about this suggestion but did not find a good way to transform the Tables into Figures. We also would like to keep Figure 1 as it presents an illustration of the field site.

2.10 The results clearly show that the networks are more stable than the null expectation. However, it should perhaps be clearly discussed that this is not surprising as the correlates of drifting (nest size, brood ratio, location), presumably stay quite stable through the observation period.

2.10 We found that result somewhat surprising because there is high turnover in nest membership, and the needs of brood care are changing constantly as larvae grow, turn into pupae, new eggs are laid, parasites appear/disappear.

2.11 Please use fewer significant figures when giving the test statistic and p-values, going down to five decimal places is not useful.

2.11 We agree. We have gone down to 2 decimals.

2.12 Remove speculative and discussive sentences from the results section, just present the results. Especially lines 369-371 – this is very speculative, what information is moved between the populations in this particular case?

2.12 We agree. We have deleted the sentence mentioned as well as lines 407-410, 420-422 and 427-430.

Intro/discussion

2.13 I don't find the "multi-level social organization" framework very helpful here (esp. in the light of my comment 4), it does not really bring much to the discussion on top of what setting this study in the framework of drifting observed in other social insects would do. In the introduction drifting due to helping and drifting due to social parasitism are discussed, and I think this suffices as a framework. If a broader framework is needed, I think comparison to polydomy in ants could be more informative.

2.13 We would like to keep the multi-level organisation framework because our study clearly demonstrates that one has to look at all these levels to understand the dynamics of between nest movement. We agree that the process of polydomy in ants is very similar. However, there has been only very little quantitative analyses of movement of workers between colonies. Also our manuscript is already quite long and a serious comparison with polydomy in ants would require quite some more additional space. But we agree that a comparison would be of interest and this is something we might be doing in the future.

2.14 the reference list has some typos and strange formattings:

Ref 10 Chaline, not Cha

Ref 16: ?

Ref 33 Sokal, not Sokhal

2.14 These typos have been corrected. Thanks for pointing them out.

Referee: 3

Comments to the Author(s)

Comments on RSPB-2020-0999

This study investigates the social structure of the paper wasp *Polistes canadensis*, using novel tracking technology and social network analyses to reveal 3 levels of social organization across which drifter wasps interact. Drifter wasps more often came from larger natal nests, moved between nests that were closer to their natal nest, and visited nests that required more help (had a higher brood to worker ratio). These results are interpreted as evidence that indirect fitness payoffs can be realised at levels beyond the nest. I thought this manuscript was well written and enjoyable to read. A considerable amount of work has obviously gone into tagging and tracking a significant number of wasps, and experimentally manipulating demand for help, and the analysis is thorough.

We thank the reviewer for this positive assessment of our work.

I do have a few, relatively minor comments on the paper, and I outline these below.

3.1 L111: Was there a chance that tagged wasps were 'double counted' on arriving at and leaving a nest? How was this dealt with?

3.1 Good point, as sometimes wasps did flit on/off the nest before settling: we avoided double counting by counting multiple registrations of any one tag within a 60s period as a single arrival/departure. We have added this into the methods.

3.2 - L123-125: Related to the point above, this line states that analyses were performed on the number of unique drifting wasps, rather than the number of drifting events. However, at other points in the manuscript, the number of drifting events is referred to (e.g. line 174 'frequency of drifting'; L257 'number of drifting events'). This is confusing so I would suggest either using consistent language throughout (if the variable is number of unique drifters), or explaining why number of drifting wasps/number of drifting events is analysed at each analyses.

3.2 See response to Rev 2.2.

3.3 - L126: When, and for how long, was the study period? Information on the years is given, but no further information on dates/durations.

3.3 This is a good point. We now provide this information in the new table requested by Reviewer 1.5 (Supplementary Table A1).

3.4 - L148: Table 3 referenced first. Can the tables be presented in the order they are referred to in the text?

3.4 This has been changed.

3.5 - L152: Does removing a subset of wasps change the dominance hierarchy? Could this explain changes in drifter wasp movements post-manipulation? I would welcome a bit more

background ecology on the species to understand things like how frequently subordinates lay eggs (this might motivate movements between non-natal nests), whether there is a queue to inherit the nest (experimentally shortening the queue in the nest next door might motivate wasps to visit more frequently if there is a chance they might attain dominance there more quickly, perhaps).

3.5 See response to Rev. 1.3. In addition, we added text to explain that removing foragers is unlikely to upset the social dynamics in this species as mortality rate for foragers is high; equally, foragers tend to be low-ranked individuals who are unlikely to supercede the queen and so the manipulation is unlikely to change any queue for dominance. Finally, it is worth noting that drifting in this species does not offer direct fitness opportunities (i.e., as social parasites or queen usurpation) (Sumner et al 2010).

3.6 - L181: I'm not clear on what a binarized version of the network is. Could you briefly explain? Equation in L198: What does k represent?

3.6 A binarized network is one where zeros and weights are converted to 0 and 1. K is another nest index. We now explain these in the text.

3.7 - L228: What is the out-degree of the nest?

3.7 This is now explained without the jargon of out-degree.

3.8 - L229: How many times?

3.8 This was run 10,000 times. This has been added to the text.

3.9 - L259: It's not clear until looking at the statistical tables that nest size and worker:brood ratio of both the donor and recipient are included in models. Can you make this explicit?

3.9 We have now added this into the Methods. Please also see response to Rev comment 2.8.

- L259: How correlated are nest size and worker:brood ratio in nests? Is there a limit to the queen's egg laying capacity such that larger nests end up with a higher worker:brood ratio than smaller nests?

3.10 To our knowledge queens in these colonies are not egg limited. All cells are full (unless the colony is in the declining phase; such nests were not used in our study), and if an egg is removed experimentally, it is usually replaced within a few minutes by the queen.

3.11 - L261: I'm not clear on why a logistic model (used for binary data) was used to analyse a count response variable (number of drifting events between pairs of nests). By L262-263, do you mean that you converted count data into 0/1 (drifting didn't occur/did occur) to analyse it logistically. This is not clear.

3.11 Thanks for pointing this out. We have rephrased.

3.12 - L305-306: I'm not sure what a trace or a directed mixing matrix is. I feel this section (L302-318) could do with a little more explaining, and I'm not clear on the analyses, or what it shows.

3.12 We've corrected this.

3.13 - Methods section in general: I think it would be clearer to subhead data analyses sections using the subheadings in the Results section. In my mind, this would help link up the different analyses to what they are testing, and enable the reader to follow through to the relevant results in the next section.

3.13 This is a good suggestion. We have applied it whenever possible.

3.14 - L329 & line 334: I'm not sure how or why these results are different. They read like they are the same thing (% drifters) but I'm not sure why the numbers are different.

3.14 This has been corrected.

Appendix B

Dear Editor

Thank you for agreeing to consider a revision of our MS. **Proceedings B RSPB-2020-1739**
As you will recall, we had made an email response to the editor to clarify a few things before considering whether we would consider resubmitting a revision to Proc B. Following a positive response from this email exchange (on 23rd sept 2020), we are pleased to resubmit our MS. We respond to the comments of the AE, Editor and the Reviewer collectively for each point, as they all pertain to the same few points.

Thanks for your consideration

The Authors

Dear Dr Sumner and co-authors,

Thank you for your letter concerning this manuscript. I have discussed the paper with the Associate Editor handling the paper, and they have responded in detail below. However to summarise, we both felt that the arguments you outlined in the letter greatly strengthened the paper, so would ask you to include those points more clearly in your revision.

The AE has given more detail below, and I have added a couple of points below theirs. But in summary, we hope you will proceed with a re-submission; the paper will go to review again, and we can't guarantee acceptance at this stage, but we are all agreed it is a very interesting question and a well-written paper, and a new version just needs to address the concerns raised in previous reviews.

With best wishes,
Loeske Kruuk
Editor

**

Comments from Loeske Kruuk (Editor)

LK1- Thank you for providing the additional statistical analyses, and that's very useful to see that they support your conclusions. However I am not following the argument as to why GLMMs are inappropriate: GLMMs do indeed allow you to account for non-independence of data, as the point of including appropriate random effects is to deal with exactly that non-independence.

LK1 -Random effects GLMMs can take account of non-independence; but correlated variables cannot be accounted for; indeed correlated explanatory variables violate the assumptions of these models and we have lots of correlated variables: nest size, number of brood, number of workers, number of drifters. Indeed, it is the very nature of these correlations that we are interested in. We now make this clear in the MS (Comment LK1). The logistic networks analyses take account of these. We explain this in the additional supplementary results that you have requested we include.

It was useful to see the GLMMs that you have run, so please just present the GLMM analyses you've described here as Supplementary Information (including full model details, what random effects have been included etc.). If you would like to include a line in the SI saying that you are doing so in response to a referee request, that would be fine; as you will

know, Proc B is trying to encourage transparency of the peer review process as much as possible.

LK2 - As requested, we now include these older (exploratory) analyses in the SI (see Additional File Text). However, please note that these were preliminary analyses and we would not normally choose to include them in this paper, as they are technically inappropriate for these type of data. We would therefore be grateful if the Editor and reviewer would consider this request carefully, as these analyses are statistically incorrect.

**

Combined comments from AE & Reviewer and Editor

1. Novelty of the study – edits to MS highlighted as ‘Comment 1’

Reviewer: This is a revision of a paper I have assessed in the first round as well. I have to say I still

struggle with its value, as the main important outcome (the adaptive significance of drifting shown by experimental manipulations) is still quite weak in its interpretation. Being comprehensive and rigorous, as well as having data that a lot of effort went into, are not to me enough for a high profile journal if the results do not provide interesting new insight, robustly supported.

To me the non-randomness of drifting is not really a novel result, as the earlier demonstration that drifting correlates with relatedness already shows non-randomness, and suggests adaptive value. Here it is shown that spatial distance explains drifting (but not simply as a function of diffusion), and is highly correlated with relatedness, and it remains unclear what is a cause and what is a consequence here, as due to lacking relatedness data spatial distance only is used. In this light, it is the variables explaining drifting patterns that are of the highest interest, and their discussion need strengthening.

AE/1: that the non-randomness of drifting is novel, when the senior author and colleagues already showed this (vis-a-vis going to genetically related nests), and the metric you use here, distance, you show to correlate with relationship. What is the big step forward here?

AE/2- the question about novelty and value of the manuscript is addressed in your letter in a clear, thorough and comprehensive way, for the first time. If you incorporate this clearly into your manuscript, in the introduction and discussion, including the critique of the earlier study by the senior author, then I believe that you have addressed this issue

As we explained in our letter of September 9, 2020, we believe that our study shows a lot more than drifting just being ‘non-random’ (see below) and we feel somewhat perplexed at the suggestion that our study lacks novelty. Much of the scientific literature (including most papers published in Proc B) is about advancing knowledge, not reporting absolute novelty or ‘firsts for science’. It is true that our study is not the first study to show that wasps drift between nests (the reviewer and AE correctly refer to a paper from one of us over 10 years ago, which first described the behaviour). That paper was the first comprehensive report of nest drifting of this kind (although it had been noted as early as 1964), and in that paper we suggested an adaptive explanation for drifting (that drifters would direct their investment to maximise need for help); but this study was purely descriptive and correlational, and very much a ‘first look’ analysis. Not wanting to criticise our own work, but Sumner et al 2007 did not employ any of the sophisticated statistical approaches and network analyses that we use here, that are required to take account of non-independent data; moreover, the 2007 study was limited to a single aggregation and lacked any manipulation experiments. As such, what we are offering in our MS to Proc B is a

completely different level of interrogation, exploration, analysis, robustness and insight into this near un-studied biological phenomenon, which has remained perplexing since W.D. Hamilton first noticed it in 1964. In our current MS we provide the most comprehensive analyses yet of drifting, using a huge dataset across several sites; we provide analytical robustness that no other study of nest-nest interactions in any social organism has ever attempted before – this has the potential to reset how researchers approach the analysis of non-independent, connected data; we then also provide a field manipulation experiment with some compelling evidence of what the adaptive value of drifting may be (please also note that these experiments are incredibly challenging to conduct in the field). There are, to date, no other papers (aside from Sumner et al 2007) published specifically on drifting in social wasps; our MS thus provides important new data, analyses and insights into an fundamental evolutionary question. If this were the first study of its kind to describe drifting and demonstrate its putative adaptive value, we would (with respect) be looking at more general journals, like *Nature* or *Science*.

We have now modified the abstract, introduction and discussion to make the novelty and value of our study more explicit.

Reviewer: the multi-level social organisation approach is interesting, but is not entirely novel (even to your own system). The senior author already showed the first two components of it in the *Current Biology* paper that, presumably, inspired the current study. That wasps move between aggregations is, perhaps, surprising, but hard to interpret as your own results show that drifting is largely driven by the relatedness/distance effect, and how cross aggregation drifting relates to this driver is unclear in the absence of relatedness data for these drifters and the nests they went to, and given that how the cross-aggregation distance relates to the scale at which you have previously identified distance/relatedness as being important is unclear. In addition, the 'anthropocene' spin you put on this, in the absence of any data on wasps in natural settings, is pure speculation.

We agree that the detection of the multi-level social organisation approach is not entirely novel, because it was eluded to in the 2007 *Current Biology* paper. But, without wanting to criticize our own work, the 2007 *Current Biology* paper was a 'back-of-the-envelope'/poor man's analysis in comparison to what we present here. We now expand on this in the introduction and Discussion. (see Comment 1 responses).

With regards to the "anthropocene" effect we agree that it remains speculative and have removed it entirely from the MS.

2. Effects of colony size edits to MS highlighted as 'Comment 2'

Reviewer: The finding that more individuals drift from larger colonies is not surprising as such, as that does not demonstrate that an individual worker is more likely to drift from a large colony (i.e. this result could be an outcome of a stable proportion of workers always drifting, which sounds less like adaptive adjustment of worker behaviour). Perhaps more could be understood if the proportions of workers drifting were analysed, if the claim is that workers adjust their behaviour to local conditions? This might strengthen the reasoning that workers make adaptive decisions.

AE/1: that your analyses of the impact of nest size on drifting look at numbers of events. It's not clear to me what your null hypothesis is here. If any given wasp has a probability of drifting (a reasonable null hypothesis?), then simply by virtue of nest size, larger nests will produce

more drifters, and vice versa for smaller nests. This is not particularly biologically interesting. You at least need to account for this, using some kind of proportional measure in your analyses, to show whether the patterns you see are simply a function of such a null model, or whether large nests produce a disproportionately higher number of drifting events than would be expected based on just size. If you have already done this, it's not clear from your methods and results section.

AE/2: Your response to the argument about nest size and statistics is convincing to me, but again this needs to be clearly incorporated into a revised manuscript so that it will be obvious to readers, and not leave them asking the same questions

This is a point that was dealt with in the discussion where we wrote: "With this caution, these experiments also suggest that drifting is not simply a probability function of group size (i.e., big nests do not donate more drifters simply because they are big): firstly, drifting rates changed in response to brood removal even when number of wasps (group size) remained unaltered; and secondly, donor nest size was not a significant explanatory variable after manipulation, even though there were fewer wasps (Table 4)." We now elaborate on this (see Comment 2, lines 528-535) and also add 'see also Supplementary file A3 for a GLMM analysis of these data).

Both the AE and reviewer propose that we analyse proportions or use GLMMs: this is far less statistically robust than the analyses we have already provided; GLMMs are not appropriate as the data are very far from independent such that there are lots of correlated variables: random effects cannot deal with this; this is the whole point of using social network analyses. We have conducted the most comprehensive and robust and sophisticated analysis one can do on these types of data; and the (new) point raised by the reviewer and the AE (above) now, were addressed in the original MS (as well as in the revised manuscript). As such, we cannot provide a better analysis of this: it is the best available and it is (we believe) a robust result. By their own admission, the Reviewer is not in fact familiar with the analytical methods we have used.

For the record, we did use as GLMMs and other 'traditional stats' as a very first look at this experiment early on in our exploration of the data. Since these analyses have now been explicitly asked for by the AE and Reviewer, we provide these informally in this letter. In all cases they support our results. But these are scientifically inappropriate analyses as all the data are dependent on each other and thus violate assumptions of GLMMs and so we would be uncomfortable presenting these in the actual MS. But, confidentially, we are happy to share these with you.

Firstly, the GLMM version of the pre-manipulation analyses (raised in Reviewer point 3): Three variables explained drifting pattern (and are in exact agreement with our new analysis in Table 3 of the MS).

- Workload (difference in worker:brood ratio between each pair of nests: i.e. the relative need for helping effort between a pair of nests) (GLMM, workload: $z = 34.01$, $p < 0.001$). This is the equivalent to the 'Worker:brood ratio donor' result in Table 3 of our MS.
- Distance between nests - drifters were more likely to visit neighbouring nests than those further away (GLMM, distance: $z = -7.16$, $p < 0.001$). This is the equivalent to the 'Distance' result in Table 3 of our MS.
- Nest size - larger nests more likely to be source nests (specialized in out-degrees) (GLMM, group size: $z = -2.58$, $p < 0.015$). This is the equivalent to the 'Donor nest-size' result in Table 3 of our MS.

Secondly, a non-network (more traditional stats – chisq/binomial tests) analysis of the manipulation experiment (i.e., of Table 4 in our MS).

- After experimental removal of wasps, wasps responded strongly in their outdegree, (visitations away from the natal nest – equivalent to ‘donor’ nests in Table 4) but not in their in-degree (visitations to non-natal nests – equivalent to ‘recipient’ nests in Table 4). We found that 12 out of the 14 nests altered their out-degree in the direction expected (i.e., decreased in-degrees) and only 2 out of the 14 nests altered their in-degree in the direction expected (i.e., increased in-degrees). Both in- and out-degree changes differ significantly from random (binomial test: out-degree, $p=0.013$; in-degree, $p=0.013$; Table below), and from each other (chi-square tests: $\chi^2=11.57$, $df=1$, $p<0.001$).

How does this compare with the results in table 4 of our MS? Outdegree indicates “donor” nests (and was significant); indegree indicates “recipient” nests, and was also significant. So, the results from these traditional stats are actually more compelling for drifters responding to need for help than those presented in our MS. But, as we explain, these methods do not take account of the non-independence of the data and this is why we prefer using a statistically more robust approach.

At the request of the Editor we now include the GLMM analyses this as a supplementary file (Additional File Text). But, given that this is statistically incorrect we ask that this is reconsidered.

3. Small effect sizes - Effects of colony size edits to MS highlighted as ‘Comment 3’

Reviewer: The main novel result is that (pre-manipulation), worker-brood ratio of both the donor and recipient explain variation in drifting. Here, however, the effect sizes are tiny, even if significant. The authors now state in the Table legends this is because of low statistical power, but isn't it the opposite: a tiny effect size, highly statistically significant sounds like a very powerful analysis, difficult to interpret biologically? Without expertise in these particular methods (the different QAP variants), it is very difficult to interpret the findings, especially since w/b ratios also highly significantly correlate with colony size. I strongly recommend additional analyses to support the conclusions here, even if I unfortunately can't offer advice on how this could be done – just seeing that the same results could be retrieved with a traditional binomial GLMM would be reassuring, even if the data points are not truly independent. Visual presentation of the results could also be helpful, but I guess such tiny effect sizes suggest that not much would be seen in scatterplots or plots of model predictions or similar.

AE: that your analyses of other drivers of drifting show inconsistent results, with very small effect sizes. It is really hard to know how much weight to put on these in terms of biological meaningfulness (also, as per the reviewer, I don't follow your argument about power and effect sizes and significance).

It is true that some of our analyses show small effect sizes. We are entirely transparent about this, and we also do not oversell the conclusions that can be made from this experiment (which appears to be the problem with regards to the concerns raised about the novelty). Strong p values and low effect sizes when a study has very large sample sizes can indeed

be difficult to interpret biologically. However, we didn't have very large sample sizes, especially for the manipulation experiment: worker:brood ratios were increased on 15 nests and decreased on 25 nests - these experiments are extremely difficult to conduct and limited by the availability of aggregations of nests, and the requirement not to manipulate too many nests within an aggregation to avoid strong perturbations. WRT to the rogue line in the legend, that was indeed incorrect and has been removed.

The reviewer and AE suggest that running GLMMs could help here; we had run these long before doing the more sophisticated statistical analyses and they provide an even stronger support. See response above for detail on these, and the added Supplementary Data. We have added in a discussion of this (see Comment 3 in revision). We hope this will now convince the reviewer that our conclusions are robust and that we are suitably acknowledging the shortcomings of our study.

4. Points raised previously by this reviewer which were not addressed (and which they have raised again) – Commented as 4.1, 4.2..etc in revision

AE: I do believe that this is a valuable manuscript, but it needs to be carefully revised bearing the points above, and the more detailed points made by the reviewer, in mind. In particular, I believe you need to re-analyse your data for the nest size effect. Finally, please make crystal clear in the revised document exactly what the steps forward are, and why they matter. There is still a greater emphasis on the size and scale of the study, as opposed to what we are learning from it that makes it of broad interest to scientists.

We now present the new analyses requested. We have highlighted clearly in our revised document what the changes are, what the caveats are in our study and what the future steps should be. See responses to Comments 1-3 above. We have also done our best to address the remaining points raised by the reviewer detailed below.

Reviewer(s)' Comments to Author:

Referee: 2

Comments to the Author(s).

4.1 The post-manipulation results repeat some, but not all of the pre-manipulation results: donor nest size and recipient nest w/b ratio lose significance (although effect sizes interestingly stay high, which also is a bit strange!). The lack of a nest size response, but the maintenance of a worker-brood ratio effect could be interpreted as workers making decisions based on the local needs. However, they don't seem to be able to direct their efforts into recipient nests according to their needs, which limits the adaptive value. This should be discussed.

This is an important point and one that we did discuss, albeit briefly, in our previous version. We now expand on this (Comment 4.1 in Revision); note this also overlaps with our responses to Comment 3 above.

4.2 the "multi-level" framework still feels unnecessary to me, it would suffice to say that "drifting is very rare between aggregates", and this would leave more space for discussing the possible adaptive features within aggregates.

We appreciate the reviewer's view on the "multi-level" framework but would like to keep it as it provides a simple and comprehensive description of the system. We hope that the reviewer will accept that there are different ways to present the system, none being right or wrong. However, we add a comment to reflect the reviewer's concerns in our recommendations for

future work, that would include more comprehensive parallel monitoring of neighbouring aggregations (Comment 4.2).

4.3 I have also have some smaller comments on the responses to my earlier comments: My comment: 2.4. It would be nice to have some indication of how common nests of the species are outside the studied locations. If there are a lot of nests not observed, it makes a difference on the interpretation of the larger spatial scale results. Of course it is not feasible to observe every single nest within flight radius and I'm not expecting that, but this should be at least briefly discussed. This is also relevant for the speculation on the effects of anthropogenic change in the discussion.

Response: 2.4 We added a test to indicate that nests are rarely found alone; they tend to be spatially discrete, in aggregations. This applies to populations in natural substrates (e.g., trees, caves) as well as anthropogenic substrates like buildings. My response: This partly answers my concern. But the main question really is then whether it is likely that there are aggregations of nests in the vicinity, or between the sampled aggregates, that were not sampled? Is it possible that there are missed aggregations where to more drifting occurs?

We add this to the discussion (see Comment 4.3 in the revised MS). The speculation on the effects of anthropogenic change are now removed, following the reviewers comments in an earlier comment (see Point 1 above).

4.4 My comment 2.7. The nests that were not sending or receiving drifters at all sound interesting, can any of their characteristics be identified?

Response 2.7 Although we did not address this explicitly, the answer is in fact in the reciprocal interpretation of the data presented: drifters are more likely to be sent from large nests and are more likely to be received by small nests. My response: this is confusing: by this logic a nest that neither sends nor receives drifters is a) small because it does not send drifters b) not small because it does not receive drifters. But maybe in the interest of space limits this is not important, if these nests did not have any particular characteristics.

We are confused by this exchange. Apologies if we added to this confusion. In response to your questions about the nests that did not send/receive drifters: 92% of nests received or sent drifters; it is not possible to draw any conclusions about those nests for which drifting was not detected as it is such a small sample size (7 nests out of the 93 monitored). Most likely it is a non-detection issue – i.e. probably all nests are involved with drifting events, but we failed to detect them during the monitoring period.

4.5 Figure 1: one suggestion towards how this could be made a bit more informative is that the location of the nests would be presented according to their spatial lay-out – at least then the effects of spatial proximity would be shown, although perhaps not in a very clear way. Still, this Figure tells very little of the results.

This figure is a 'high-level' overview of the methods and the study system, and not intended as a detailed depiction of results. Fig 1 gives a visual overview of the wasps, their nests, how they are situated in aggregations and how this reflects the interactions at the different levels. We have already moved the body of text which refers to Figure 1 from the Methods to the Results, on the recommendations of another reviewer. We could move it back, to make it clear that Figure 1 is not a result, but is instead aiming to help explain the set up/study system/methods/sample sizes/structure of the study. (See Comment 4.5 in Revision).

Appendix C

MULTI-LEVEL SOCIAL ORGANISATION AND NEST-DRIFTING BEHAVIOUR IN A EUSOCIAL INSECT

Thibault Lengronne^{1,2,*}, David Mlynski^{4,*}, Solenn Patalano^{2,5}, Richard James^{3,#}, Laurent Keller^{1,#} and Seirian Sumner^{2,6,#}

DOI: 10.1098/rspb.2021-0275

Electronic Supplementary Materials

This document contains:

- **Additional File Table A1:** Details for each of the three study periods (in 2005, 2009, and 2010) on sample sizes (number of aggregations, nests and tagged wasps), and data used in each of the analyses.
- **Additional File Figure A1: Drifting is persistent over time.** Social networks of drifting events between nests from the 2005 aggregations for each of the 4 monitoring periods (of 5 days each). Edges in red represent persistent drifting events over time (edges observed in at least two monitoring periods). Black edges are drifting events observed only on 1 monitoring period (edges non-persistent over time). In all four figures, nodes (representing nests) are organized according to their geographical coordinates.
- **Additional File Text: Exploratory statistics**

Additional File Table A1

Details for each of the three study periods (in 2005, 2009, and 2010) on sample sizes (number of aggregations, nests and tagged wasps), and data used in each of the analyses.

	2005	2009	2010
RFID monitoring period	16 June - 13 July	17 June - 21 July	6 June - 17 July
No. wasps tagged (incl. retagging)	422	663	619
No. nests	33	32	28
No. aggregations	2	4	2
Data used in general network patterns	y	y	y
Data used in testing randomness of networks	y	y	y
Data used for biological traits to explain drifting networks	n	y	y
Manipulation experiment	n	y	y

Additional File Figure A1: Drifting is persistent over time.

Social networks of drifting events between nests from the 2005 aggregations for each of the 4 monitoring periods. Edges in red represent persistent drifting events over time (edges observed in at least two monitoring periods). Black edges are drifting events observed only on 1 monitoring period (edges non-persistent over time). In all four figures, nodes (representing nests) are organized according to their geographical coordinates.

PERIOD 1

PERIOD 2

PERIOD 3

PERIOD 4

Additional File: Exploratory Statistics

GLMM analysis of the pre-manipulation analyses. In the data exploration phase, we conducted GLMM analyses to explore the possible variables that explain drifting patterns. At the request of the reviewer, we include a GLMM analysis of the data. Three variables explained drifting pattern (and are in exact agreement with the analysis presented in Table 3 of the paper).

In the first model we used number of drifting events between pairs of nests ($n=87$ nests) as a response variable, with relatedness, distance (between nests, in meters) and workload (estimated as the difference in worker:brood ratio between each pair of nests (i.e. (brood number of nestA * the group-size of nestB) / (brood number of nestB * the group-size of nestA)) as explanatory variables, and “site” (aggregation) and “time” (year of experiment) as random factors. Drifters were more likely to visit nests with low worker:brood ratios (GLMM, workload: $z= 34.01$, $p<0.001$). This is the equivalent to the ‘Worker:brood ratio donor’ result in Table 3 of the paper. Drifters were more likely to visit neighbouring nests than those further away (GLMM, distance: $z=-7.16$, $p<0.001$). This is the equivalent to the ‘Distance’ result in Table 3 of the paper.

We performed a separate GLMM to explore other effects, including group size. We used net in- and out-degree per nest (in-degrees minus out-degrees) as the response variable, group size and the presence of brood parasites as explanatory variables, and “site” (aggregation) and “time” (year of experiment) as random factors. This revealed that larger nests were more likely to be source nests for drifters (specialized in out-degrees) (GLMM, group size: $z= -2.58$, $p<0.015$). This is the equivalent to the ‘Donor nest-size’ result in Table 3 of the paper. There was no significant effect of parasite presence.

Exploratory statistics (chisq/binomial tests) of the manipulation experiment (i.e., of Table 4 in the paper) After experimental removal of wasps, wasps respond strongly in their outdegree, (visitations away from the natal nest – equivalent to ‘donor’ nests in Table 4) but not in their in-degree (visitations to non-natal nests – equivalent to ‘recipient’ nests in Table 4). We found that 12 out of the 14 nests altered their out-degree in the direction expected (i.e. decreased in-degrees) and only 2 out of the 14 nests altered their in-degree in the direction expected (i.e. increased in-degrees). Both in- and out degree changes differ significantly from random (binomial test: out-degree, $p=0.013$; in-degree, $p=0.013$), and from each other (chi-square tests: $\chi^2=11.57$, $df=1$, $p<0.001$). In relation to the analysis presented in Table 4, outdegree indicates “donor” nests, and indegree indicates “recipient” nests. Both suggest a significant response to the manipulation.